# What MLLMs Learn about When they Learn about Multimodal Reasoning: Perception, Reasoning, or their Integration?

## Abstract

Multimodal reasoning models have recently shown promise on challenging domains such as olympiad-level geometry, yet their evaluation remains dominated by aggregate accuracy, a single score that obscures where and how models are improving. We introduce MathLens, a benchmark designed to disentangle the subskills of multimodal reasoning while preserving the complexity of textbook-style geometry problems. The benchmark separates performance into three components: *Perception*: extracting information from raw inputs, *Reasoning*: operating on available information, and *Integration*: selecting relevant perceptual evidence and applying it within reasoning. To support each test, we provide annotations: visual diagrams, textual descriptions to evaluate reasoning in isolation, controlled questions that require both modalities, and probes for fine-grained perceptual skills, all derived from symbolic specifications of the problems to ensure consistency and robustness. Our analysis reveals that different training approaches have uneven effects: First, reinforcement learning chiefly strengthens perception, especially when supported by textual supervision, while textual SFT indirectly improves perception through reflective reasoning. Second, reasoning improves only in tandem with perception. Third, integration remains the weakest capacity, with residual errors concentrated there once other skills advance. Finally, robustness diverges: RL improves consistency under diagram variation, whereas multimodal SFT reduces it through overfitting. We will release all data and experimental logs.

## 1 Introduction

Recent advances in reasoning with Large Language Models (LLMs) have yielded remarkable progress in challenging domains such as Olympiad-level mathematics (Mathematical Association of America (2025)), graduate-level scientific question answering (Rein et al. (2024)), and multi-step program synthesis (Austin et al. (2021); Chen et al. (2021)). Motivated by these successes, a natural extension is to adapt similar training paradigms to Multimodal Large Language Models (MLLMs), equipping them with reasoning capabilities over both text and visual inputs. Tasks such as mathematical problem solving (Lu et al. (2024); Wang et al. (2024b)) and visual puzzles (Dao & Vu (2025); Ghosal et al. (2024); Feng et al. (2025)) illustrate the potential of this direction, giving rise to methods that adapt Supervised FineTuning (SFT) (Sun et al. (2025); Chung et al. (2025)) and Reinforcement Learning (RL) (Deng et al. (2025); Meng et al. (2025)) for multimodal reasoning, including sequentially combining both stages for enhanced reasoning.

However, unlike LLMs where reasoning-oriented training consistently yields substantial gains, multimodal reasoning training exhibits highly variable outcomes. This motivates analysis of how different training strategies influence specific skills. To this end, multimodal reasoning may be decomposed into perception, reasoning, and their integration, as each corresponds to a distinct source of error. Existing benchmarks, however, rarely adopt such a decomposition and instead primarily report aggregate accuracy, which obscures these distinctions. Some vary input modalities to approximate skill-specific testing, but without strict controls they fail to isolate capacities and offer limited diagnostic value (see Appendix B). Consequently, it remains unclear when multimodal reasoning training benefits MLLMs and which training signals or architectural choices are responsible.

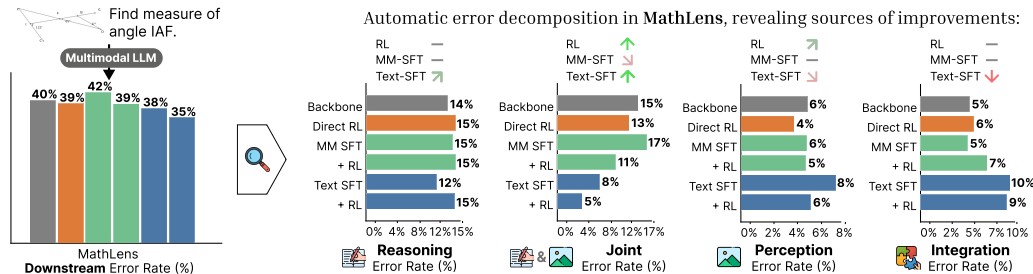

Figure 1: MATHLENS decomposes multimodal reasoning errors into perception, reasoning, and integration, revealing capacity-specific shifts after fine-tuning that are hidden by aggregate accuracy. Each training strategy affects capacities differently; e.g., text SFT yields a minor gain (↗) in reasoning but harms (↓) integration (model details in appendix D).

To close this gap, we present MATHLENS, a controlled benchmark of geometry problems that isolates perception, reasoning, and their integration (fig. 1). Starting from the symbolic semantic state of (Zhang et al., 2023), we build four aligned annotations (section 2.2): (i) diagrams rendered from geometric constraints to test perception, (ii) a definitionally equivalent textual description to test reasoning when perception is trivialized, (iii) multimodal questions requiring both modalities, and (iv) fine-grained probes targeting recovery of visual details. To further test robustness against visual modifications, MATHLENS introduces semantic diagram modifications that alter visual form while preserving task correctness. By grounding the benchmark in geometry, MATHLENS retains authentic task complexity and enables rigorous comparison to prior benchmarks.

MATHLENS demonstrates that training strategies shape multimodal reasoning capacities in distinct ways. 1) *Perception* is primarily boosted by reinforcement learning, with larger gains when strong textual SFT has already established reasoning competence (section 3.2), while textual SFT itself, despite lacking visual input, indirectly strengthens perception through reflective reasoning (section 3.3). 2) *Reasoning* improves in tandem with perception under RL, but does not exhibit distinct additional gains beyond those coupled improvements (section 3.4). 3) *Integration* remains the least improved capacity: RL offers little benefit, and as perception and reasoning advance together, residual errors increasingly concentrate in integration, leaving it as the dominant failure mode (section 3.4). 4) *Robustness* diverges across strategies, with RL enhancing consistency under diagram variation, whereas multimodal SFT reduces robustness through overfitting (section 3.5).

Our contributions are:

- **Framework for multi-axial evaluation of multimodal reasoning:** A new benchmark that disentangles perception, reasoning, and integration, enabling analysis beyond aggregate accuracy.
- **Findings from controlled analysis:** Discoveries about how different training objectives and data settings influence the capabilities of Multimodal Language Models.
- **MATHLENS dataset:** A public release of all problems, annotations, evaluation scripts, and perturbation generators, establishing a reproducible resource for the community.

## 2 MATHLENS

Most multimodal reasoning benchmarks report only aggregate accuracy, obscuring whether errors stem from perception (extracting information from inputs), reasoning (operating on extracted information), or their interaction. Our benchmark is designed to *disentangle these subskills* while preserving realistic problem contexts. MATHLENS comprises 926 geometry problems, each presented with eight visual modifications and an average of $\sim 7.03$ visual probes per problem.

### 2.1 DATASET FORMALIZATION

**Definitions.** For a problem instance $k$, we consider two latent generative variables: the context semantics $S_k$ and the query operator $\varphi_k$. The semantics decomposes atomically,

$$S_k = \{s_{k,1}, \ldots, s_{k,m}\},$$

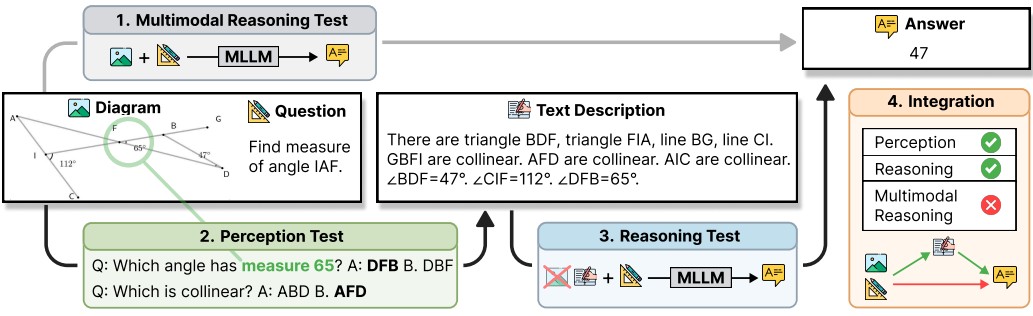

Figure 2: Based on MATHLENS annotations, the joint *Multimodal Reasoning Test* (1) is decomposed into a Perception Test and a textual Reasoning Test. The *Perception Test* evaluates questions answerable directly from the diagram, such as reading an annotated angle (e.g., $\angle DFB$ in (2)). The *Reasoning Test* (3) replaces the diagram with a complete textual description (e.g., "There are triangle $BDF \ldots \angle DFB = 65°$"), such that the question can be solved without visual access. Finally, *Integration* (4) highlights cases where multimodal reasoning fails even though perception and reasoning, when tested independently, succeed.

where each $s_{k,i}$ encodes a basic fact. For example, in a geometry problem $k$, $s_{k,1}$ could represent ($\angle ABC = 50°$). Both visual diagram $C_k^{img}$ and textual description $C_k^{txt}$ context are generated conditionally on $S_k$,

$$C_k^{\text{img}} \sim p(C^{\text{img}} \mid S_k), \qquad C_k^{\text{txt}} \sim p(C^{\text{txt}} \mid S_k).$$

The surface question $Q_k$ is generated from $\varphi_k$ and a subset of atoms,

$$Q_k \sim p\big(Q \mid \varphi_k, S_k^{\text{q}}\big), \qquad S_k^{\text{q}} \subseteq S_k,$$

with $\varphi_k = (\text{COMPUTE } \angle A)$ and $S_k^{\text{q}} = \{s_{k,1}\}$, the question can be "$\angle ABC = 50°$, what is $\angle A$?" The ground-truth answer is defined by applying the operator to the full semantic state,

$$A_k = f(\varphi_k, S_k).$$

Further, we generate a perception probe set $Q_k^{\text{perc}}$ with semantic atoms $s_{k,i}$. Let

$$Q_k^{\text{perc}} = \{q_{k,i}^{\text{perc}}\}_{i=1}^m, \qquad a_{k,i}^{\text{perc}} = [\![s_{k,i}]\!],$$

so each probe targets a single atom of $S_k$ and its gold answer is the atom's valuation. For example, $q_{k,1}^{\text{perc}} = $ "Which angle has measure of 50? A. ABC B. ABD" with $a_{k,1}^{\text{perc}} = $ A. These probes directly test whether a model has recovered the components of $S_k$ from the observed context.

**Isolating subskills with $S_k$ access.** Having access to the semantic state $S_k$ allows us to design tests that separate perception, reasoning, and their integration, as shown in Figure 2.

*Perception.* Probe questions $Q^{\text{perc}}k$ with gold answers $a^{\text{perc}}k, i$ are derived from the atomic facts listed in $S_k$. These probes check if the model can recover the specific facts in $S_k$ that are needed to solve the problem from the given input. Errors here indicate pure perceptual failures.

*Reasoning.* Given $S_k$, we render a textual description $C_k^{\text{txt}}$ that directly encodes the relevant details. Evaluating on $(C_k^{\text{txt}}, Q_k)$ trivializes perception, so the task reduces to applying $\varphi_k$ correctly. Errors here isolate reasoning competence, free from perceptual confounds.

*Integration.* We isolate integration effects by combining accuracy on $(C_k^{\text{img}}, Q_k)$ with auxiliary perception and reasoning measures. Conditional on success in perception probes and text-only reasoning, any remaining errors on the full task are treated as integration errors; this defines integration as the residual after controlling for perception and reasoning under standard coverage assumptions.

---

[1]In this example, an obtuse angle is drawn acute. Such non-canonical diagrams block visual estimation and enforce geometric deduction, consistent with Olympiad practice ("not drawn to scale," e.g., AMC).

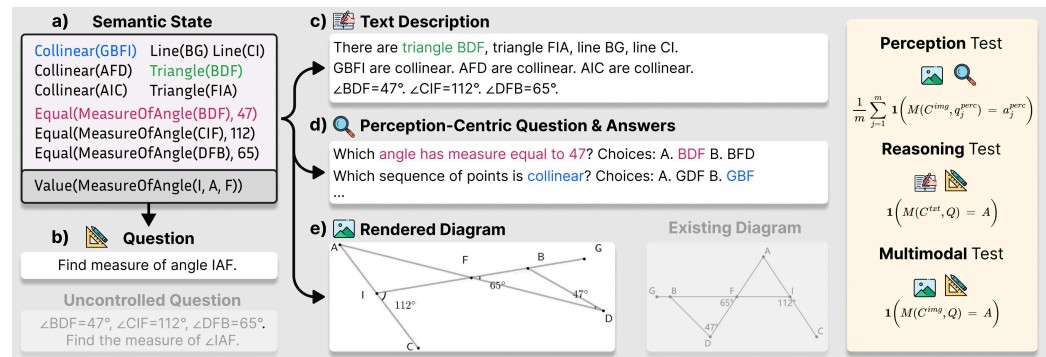

Figure 3: **Sample data generation process in MATHLENS.** From a *semantic state* representation, we build controlled *text descriptions*, *perception probes*, and *questions* with no overlap with visual content. Also, new *diagrams* are rendered from the semantic state to avoid visual familiarity effects[1].

**Additional uses of $S_k$.** First, $S_k$ allows construction of questions that *require context*. We define $Q'_k$ by restricting its atoms to exclude those in the context latent $S^c_k$:

$$Q'_k \sim p(Q \mid \varphi_k, S^{q'}_k), \qquad S^{q'}_k \subseteq S_k, \qquad S^{q'}_k \cap S^c_k = \varnothing.$$

By design, $Q'_k$ cannot be answered from the query alone and thus forces reliance on $C_k$.

Second, the symbolic representation $S_k$ allows us to apply *semantic perturbations* of the context. Let $\tau \in \mathcal{T}_{AP}$ denote a transformation from the set of admissible perturbations (e.g., relabeling points, rotating a diagram, or permuting equivalent elements). Applying $\tau$ yields a perturbed specification $S'_k = \tau(S_k)$, while preserving the problem semantics so that

$$f(\varphi_k, S_k) = f(\varphi_k, S'_k).$$

Re-rendered contexts ($C^{img}_{k'} \sim p(C^{img} \mid S'_k)$ and $C^{txt}_{k'} \sim p(C^{txt} \mid S'_k)$) serve as systematic distractors. We measure robustness by prediction agreement across contexts from $S_k$ and $S'_k$, testing semantic invariance beyond pixel-level augmentation. Pixel-level perturbations are insufficient, since they often cause prediction shifts driven by abnormal appearance rather than semantic change.

## 2.2 DATA GENERATION PIPELINE

Thus, we build on latent semantics $S_k$ for a set of practical multimodal reasoning problems, matching problem domain and complexity with popular benchmarks in the literature (Zhang et al. (2024); Lu et al. (2024)). Each instance is first specified symbolically as $S_k$ and $\varphi_k$, then rendered into its observable forms: diagrams $C^{img}_k$, textual descriptions $C^{txt}_k$, and multiple types of questions.

**Data source (fig. 4 a).** We build on FormalGeo-7K (Zhang et al. (2023)), which provides symbolic annotations for geometry problems. Each diagram cue or condition is encoded as a predicate (e.g., COLLINEAR($AB, BC$), EQUALLENGTH($AB, CD$)), forming the semantic state $S_k$. This yields realistic problems with formal representations from which we build the required artifacts.

**Questions (fig. 4 b).** We represent each question $Q_k$ as clauses $[S^q_k; \varphi_k]$, with facts $S^q_k$ and goal operator $\varphi_k$. For a strictly multimodal question, we drop clauses overlapping with the context $S^c_k$:

$$Q'_k = f_q\big([(S^q_k \setminus S^c_k); \varphi_k]\big),$$

where $f_q$ linearizes them into natural language. Thus $Q'_k$ requires contextual information to solve.

**Textual descriptions (fig. 4 c).** Each detail $s_{k,i} \in S_k$ is mapped by template function $f_d$ to clause,

$$C^{txt}_k = \text{concat}_i f_d(s_{k,i}),$$

yielding a faithful textual rendering of formal representation $S_k$ without stylistic variation (e.g., EQUALLENGTH($AB, CD$)→ "Segment $AB$ is equal in length to segment $CD$").

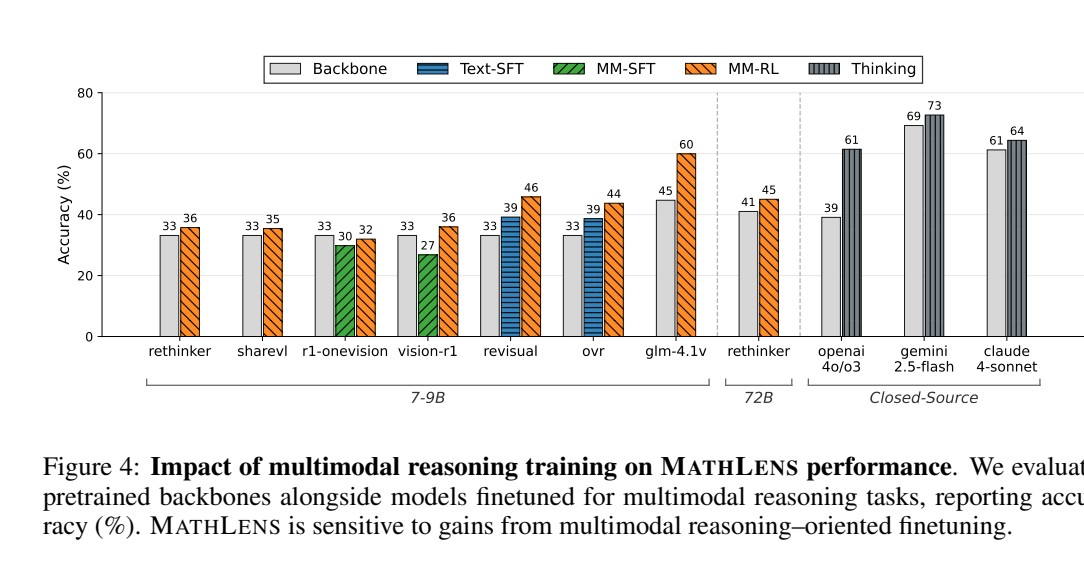

Figure 4: **Impact of multimodal reasoning training on MATHLENS performance**. We evaluate pretrained backbones alongside models finetuned for multimodal reasoning tasks, reporting accuracy (%). MATHLENS is sensitive to gains from multimodal reasoning–oriented finetuning.

**Perception Probes (fig. 4 d).** Each atomic detail $s_{k,i} \in S_k$ is converted to a probe via templating function $f_{\mathrm{p}}$ with gold answer $[\![s_{k,i}]\!]$:

$$Q_k^{\mathrm{perc}} = \{f_{\mathrm{p}}(s_{k,i})\}_i, \qquad a_{k,i}^{\mathrm{perc}} = [\![s_{k,i}]\!].$$

For example, COLLINEAR$(AB, BC)$ yields "Which points are collinear? A. ABC B. ABD" with answer A and negative B. Thus probes directly test recovery of $S_k$ from context.

**Diagram rendering (fig. 4 e).** Each clause $s_{k,i} \in S_k$ is converted into geometric constraints (e.g., PERPENDICULAR$(AB, BC) \rightarrow (x_a - x_b)(x_c - x_b) + (y_a - y_b)(y_c - y_b) = 0$). A numerical solver computes coordinates that satisfy all constraints, which are then rendered into diagrams. We then manually filter outputs to remove artifacts such as occlusions or overlaps.

**Diagram modifications.** To test robustness against distractions, we alter the semantic state $S_k$ to generate diagram variants. Modifications include adding auxiliary geometric elements, applying flips or rotations, merging instances, and relabeling points, while preserving the ground-truth answer. All variants are manually screened to discard visually invalid cases.

### 2.3 MATHLENS-GENERAL: A COMPLEMENTARY GENERAL-DOMAIN SET

The main dataset, MATHLENS, solely consists of geometry problems. While this symbolic structure enables precise control, it also risks overfitting analyses to a narrow domain. To mitigate this limitation, we construct MATHLENS-GENERAL, a set of multimodal reasoning problems spanning *diverse domains*. Unlike MATHLENS, it cannot maintain the same rigor afforded by formal semantic states, but is instead curated from prior sources through a rigorous pipeline to ensure reliability and diversity. Curation procedures and experimental results are provided in Appendices C and E.

## 3 EXPERIMENTS

We focus on open-weight multimodal reasoning models in the 7–9B parameter range, along with the backbone counterparts. This setup is motivated by two factors: transparency of training methods and data, which is available only for open models, and the fact that most publicly released multimodal reasoning models fall within this size range. We assemble seven model families, yielding 13 checkpoints in total by including both SFT-only and RL-finetuned variants where available. We include Qwen-2.5-VL (Bai et al., 2025) and GLM-4.1V-Base (Hong et al., 2025) as backbone models. Direct multimodal Reinforcement Learning (RL) models comprise VL-Rethinker (Wang et al., 2025) and ShareVL-R1 (Yao et al., 2025). Models trained with multimodal Supervised Finetuning (SFT) followed by multimodal RL include Vision-R1 (Huang et al., 2025) and R1-Onevision (Yang et al., 2025), while those using textual SFT followed by multimodal RL include Revisual-R1 (Chen et al.,

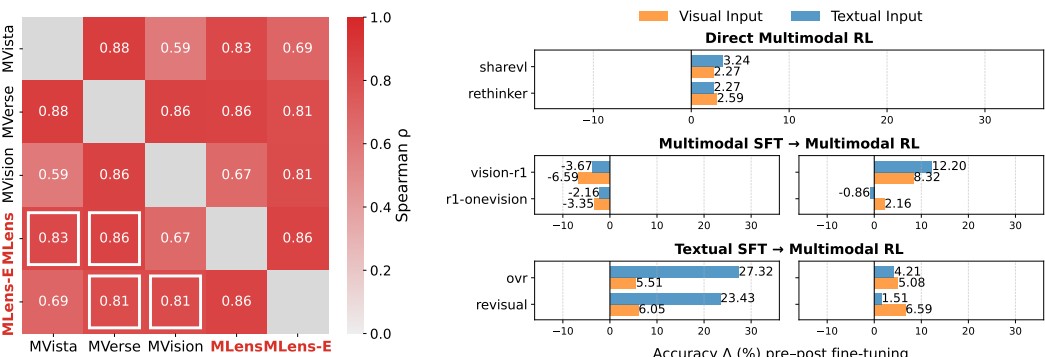

Figure 5: (**left**) **Correlation of MATHLENS with popular benchmarks.** MATHLENS shows high correlation with standard multimodal reasoning benchmarks. (**right**) **Performance gains by input modality.** Bars show percentage point shifts from finetuning for text versus diagram inputs. Visual gains exceed textual ones when models are primed with strong reasoning (textual SFT).

2025)[2] and Open-Vision-Reasoner (OVR) (Wei et al., 2025). Finally, GLM-4.1V-Thinking (Hong et al., 2025) falls outside these categories, since its training data is not publicly disclosed.

We additionally evaluate six larger models, yielding eight runs in total by including both Gemini-2.5-Flash (Comanici et al., 2025) and Claude 4 Sonnet (Anthropic, 2025) in "thinking" and "non-thinking" modes: Qwen-2.5-VL-72B, VL-Rethinker-72B, Gemini-2.5-Flash, Claude 4 Sonnet, and GPT-O3/4O (OpenAI, 2025). For each model, we follow the recommended decoding configurations to generate both reasoning and final answers. Most models employ greedy decoding and yield deterministic outputs. Full experimental details and results are in Appendix E.4.

**Weak textual reasoning in existing multimodal SFT models.** We take it as baseline that current multimodal SFT models show limited reasoning following Chen et al. (2025): their data are easier than text-only sets, leading to weaker performance on reasoning benchmarks. Building stronger data is hindered by the lack of open multimodal reasoning traces (see Appendix E.1 for discussion).

### 3.1 VALIDATING MATHLENS AS A MULTIMODAL REASONING BENCHMARK

Before using MATHLENS to study finetuning effects, we first validate it as a multimodal reasoning benchmark, showing that it captures the same skills as established benchmarks.

**Sensitivity to finetuning.** We assess whether MATHLENS reflects gains from multimodal reasoning finetuning by comparing pretrained backbones with their finetuned variants on its downstream task of solving geometry problems from diagrams. As shown in Figure 4, finetuned models consistently surpass backbones, demonstrating that MATHLENS is sensitive to these adaptations.

**Correlation with established benchmarks.** To test whether MATHLENS captures patterns consistent with prior benchmarks, we compare it to MathVista (Lu et al. (2024)), MathVerse (Zhang et al. (2024)), and MathVision (Wang et al. (2024b)) by correlating model accuracies across eight models (Figure 5, left; full results in Appendix). MATHLENS shows strong Spearman's $\rho$ correlation with MathVista ($\rho = 0.83$) and MathVerse ($\rho = 0.86$), confirming alignment with established benchmarks, while correlation with MathVision is weaker but still positive ($\rho = 0.67$). Importantly, our goal is not a new downstream benchmark but a controlled, decomposition-focused resource; high correlations therefore underscore its consistency.

### 3.2 HOW MUCH OF THE GAIN IS EXPLAINED BY IMPROVEMENT IN TEXTUAL REASONING?

To isolate the role of textual reasoning, we compare performance on textual descriptions $C_k^{\text{txt}}$ and visual diagrams $C_k^{\text{img}}$, which encode identical information conditioned on the question $Q_k$. This

---

[2]Revisual-R1 adds additional textual RL, yet we group it as textual SFT → multimodal RL for consistency.

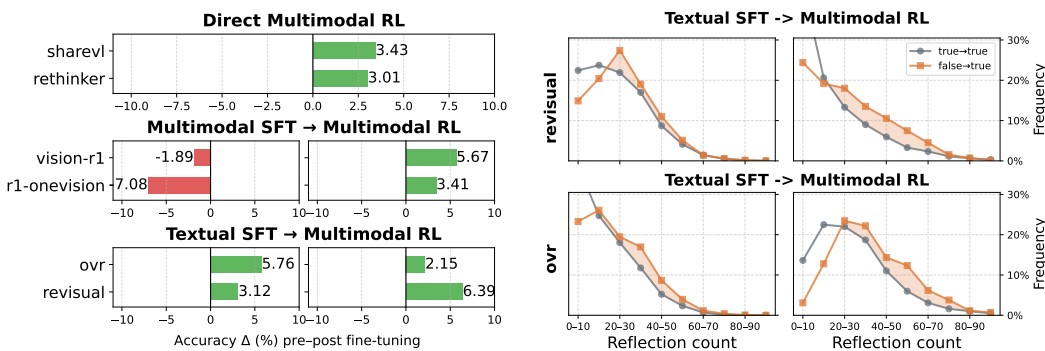

Figure 6: (**left**) **Perception performance shifts from finetuning** in percentage points. Except for multimodal SFT, all methods contribute to increased perception capacity. (**right**) **Sample distribution over reflection count.** Reflection is more frequent in false→true cases than in true→true cases, showing that perception gains from textual SFT arise partially from reflective reasoning.

parallel annotation lets us track how multimodal training affects each modality. We report accuracy differences (percentage points) before and after multimodal finetuning.

Figure 5 (right) shows that textual SFT mainly improves textual reasoning, while multimodal RL applied afterward yields larger multimodal gains by shifting its effect to perception and integration. Direct multimodal RL without textual SFT gives only modest improvements in both modalities, and poor-quality multimodal SFT degrades multimodal performance more than textual reasoning, with RL only partially recovering the gap.

> Finding 1: **Multimodal RL impact varies with textual reasoning strength.** With strong textual SFT it mainly boosts perception; without it, it modestly improves both modalities (fig. 5).

### 3.3 How much of the gain is explained by improvement in perception?

Next, we evaluate how multimodal reasoning training affects low-level perception. For each problem $Q_k$, we use the perception probes $Q_k^{\text{perc}} = q_{k,1}^{\text{perc}}, \ldots, q_{k,j}^{\text{perc}}$. Figure 6 (left) shows that all RL models contribute to better perception required for geometry problem solving. Hence, we conclude that perception is elicited even by training with correctness reward signals on the downstream problems.

**How textual SFT improves perception.** Textual SFT does not use visual inputs, yet it enhances perceptual performance. These gains suggest influences beyond direct perceptual learning. We hypothesize that one contributing factor is that enhanced reasoning alters how the model interprets ambiguous visual evidence. In particular, stronger reasoning promotes cognitive strategies such as *reflection* and self-correction, which allow the model to revise initial perceptual judgments.

We examined the reasoning traces of two models with textual SFT in Figure 3 (right). Correct predictions were divided into those accurate both before and after training (true→true) and those that became accurate only after training (false→true). We found that reflective reasoning was more frequent in the latter, where accuracy improved post-training. These results indicate that textual SFT promotes reflective reasoning, which enables models to revisit and correct initial perceptual errors.

> Finding 2: **Textual SFT also improves perception** by fostering reflective reasoning (fig. 6).

### 3.4 Error type analysis

MATHLENS's annotation designs enable systemic categorization of model outputs into the following error categories: (1) *Perception & Reasoning*, failures on both perception probes and textual reasoning; (2) *Perception*, failure on perception probes but correct reasoning from text; (3) *Reasoning*, correct perception probes but failed text-based reasoning; (4) *Integration*, correct perception

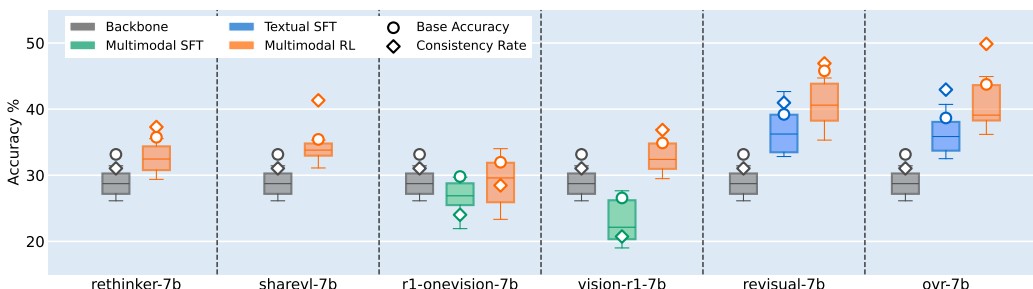

Figure 8: **Robustness to semantic-level visual modifications.** Box plots show accuracy on modified diagrams. Points report accuracy on the unmodified base diagrams and the overall consistency score. Multimodal RL improves consistency, whereas multimodal SFT reduces it.

and text in isolation but failure on the combined multimodal task. Correct categories: (5) *Trivial*, solvable from text alone; and (6) *Rest*, all other correct cases.

Figure 1 (see full results in Figure 15 of Appendix) shows that RL primarily reduces perception and reasoning errors on the same problems, indicating correlated gains. Yet many of the reductions manifest as *Integration* errors, suggesting that finetuning often shifts them into coordination failures.

> Finding 3: **RL improves perception and reasoning in a correlated manner, and unveils integration as the remaining source of error** once the other capacities improve (fig. 1).

### 3.5 DOES MULTI-MODAL FINETUNING EFFECT MODEL'S ROBUSTNESS TO VISUAL INPUTS?

We extend the visual familiarity analysis from Section 3.1 by testing robustness under controlled diagram variations. Instead of pixel-level augmentations (e.g., blurring), we apply *semantic* modifications to the geometric specification, isolating structural variation without introducing low-level artifacts (see Appendix C).

**Downstream accuracy can mask familiarity effects.** Figure 7 shows that while most training methods are stable under familiar diagrams, multimodal SFT suffers a sharp drop on modified diagrams. This indicates that similar downstream accuracy on public benchmarks can hide reliance on visual familiarity.

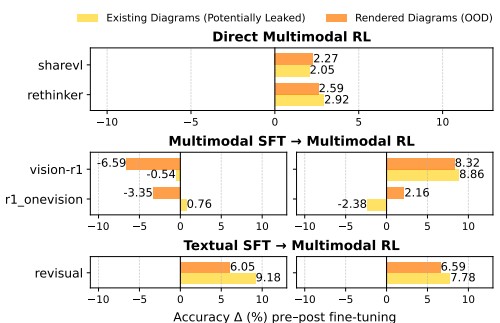

Figure 7: **Finetuning shifts by diagram type.** Bars show percentage point changes for existing versus out-of-distribution rendered diagrams.

**RL improves visual consistency.** To quantify robustness, we use Consistency Rate (CR) (Zhao et al. (2024)), the expected agreement of predictions across perturbations of the same diagram:

$$CR = \mathbb{E}_{Q'_k \sim Q'} \left[ \mathbb{E}_{\substack{\bar{C}_{k,i}, \bar{C}_{k,j} \sim \bar{C}_k \\ i \neq j}} \mathbf{1}\big[M(Q'_k, \bar{C}_{k,i}) = M(Q'_k, \bar{C}_{k,j})\big] \right], \tag{1}$$

where $\bar{C}_k$ is the set of diagram variants for question $Q'_k$.

Figure 8 shows that accuracy on base diagrams is generally higher than on modified ones, indicating vulnerability to semantic perturbations. Consistency increases after multimodal RL, suggesting robustness to variation. By contrast, multimodal SFT lowers consistency.

> Finding 4: **Multimodal RL improves visual consistency** under structural variations (fig. 8).

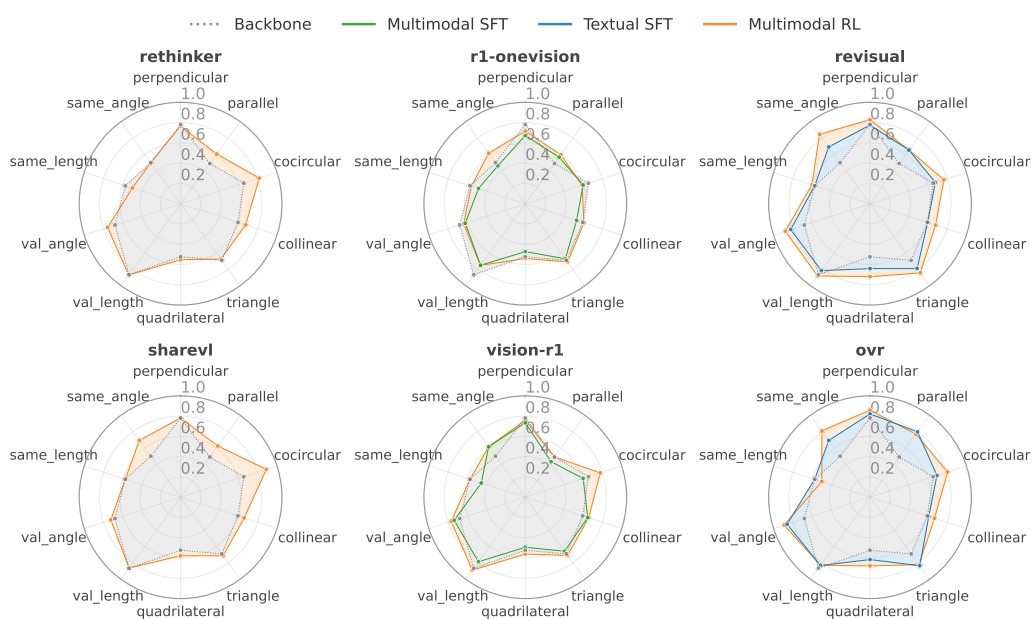

Figure 9: **Perception probe results by question type.** Improvements differ across skills: angle-related and cocircularity tasks improve with multimodal RL, line-length tasks show little change, and polygon detection improves only with textual SFT. A likely factor is whether notations are colocated with the geometric element they describe or spatially separated.

### 3.6 WHICH PERCEPTION SKILLS BENEFIT FROM MULTIMODAL REASONING TRAINING?

We decompose perception probe performance (Section 3.3) by relation type (full details in Appendix E.4). This analysis reveals uneven gains, as shown in Figure 9. Relation types such as *cocircular*, *parallel*, and *collinear* improve consistently, as they rely on simple primitives. *same_angle*, *val_angle*, *triangle*, and *quadrilateral* improve only with textual SFT plus multimodal RL, since they require multi-constraint reasoning or symbol–geometry links. By contrast, *same_length*, *val_length*, and *perpendicular* remain difficult, as their cues are spatially offset or visually ambiguous.

> Finding 5: **Perception gains are uneven.** Direct geometric cues improve reliably, while tasks relying on symbolic marks or distant annotations remain difficult (fig. 9).

## 4 CONCLUSION

We introduced MATHLENS, a controlled benchmark that disentangles perception, reasoning, integration, and robustness in multimodal reasoning. Our findings show that reinforcement learning primarily boosts perception, with stronger gains when supported by textual supervision, while textual SFT indirectly strengthens perception through reflective reasoning. Reasoning improves in tandem with perception under RL but does not exhibit distinct additional gains, leaving integration as the least improved capacity and the dominant failure mode once other skills advance. Robustness further diverges across strategies, as RL enhances consistency under diagram variation, whereas multimodal SFT reduces it through overfitting.

Looking ahead, our results motivate future architectures and training strategies that explicitly target integration, for example by introducing auxiliary pretext objectives for RL that enforce cross-modal grounding, or by structuring training data to better capture causal correspondences between perceptual details and reasoning trajectories. In parallel, scaling atomic perception probes into auxiliary supervision offers a promising direction for directly improving perceptual capacity.

## ETHICS STATEMENT

MATHLENS consists solely of mathematical problems and does not involve human subjects or sensitive data. MATHLENS-GENERAL is constructed as an extension of existing benchmarks, and ethical considerations are therefore inherited from those sources. All human annotations were performed directly by the authors. As the benchmark is derived from publicly available resources, it does not raise additional privacy or copyright concerns beyond those already addressed in the original sources. As the content is limited to mathematical problems, concerns of bias, fairness, or harmful applications are not applicable.

## REPRODUCIBILITY STATEMENT

All experiments are conducted using existing models without additional training. The complete list of models is provided in Table 1, with hyperparameter configurations in Appendix D. Results are reported under deterministic decoding, aside from a small subset of models requiring random sampling to mitigate repetition. Minor nondeterminism from computation kernels, common across current LLM decoding environments (both local and API-based), is generally not treated as a controlled factor (He & Lab, 2025). As API services may evolve over time with undocumented changes, our main experiments focus on open-weight models, with API model results reported as auxiliary reference. The full dataset and model outputs will be released for reproducibility.

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

OVERVIEW OF THE APPENDIX

This Appendix is organized as follows:

- Appendix A contains related work;
- Appendix B contains a comparison with other benchmarks and discusses the benefit of automatic error analysis;
- Appendix C contains details of the data generation process for the main MATHLENS subset as well as the MATHLENS-GENERAL subset;
- Appendix D contains implementation details, including hyperparameters and computational resources;
- Appendix E contains the complete experimental results corresponding to the main paper figures, as well as results on MATHLENS-GENERAL;
- Appendix F contains qualitative examples.

## A   RELATED WORK

**Evaluation of multimodal reasoning capacity.**   Multimodal reasoning is a compositional process that requires the integration of perception and abstract reasoning (Li et al. (2025)). Inspired by textual reasoning benchmarks, recent multimodal reasoning datasets focus on verifiable domains such as mathematics (Lu et al. (2024); Wang et al. (2024b); Zhang et al. (2024)), scientific diagrams (Yue et al. (2024a); Hao et al. (2025)), and charts (Wang et al. (2024c)). However, most of these benchmarks report only a single downstream accuracy per model, without systematic means to identify the source of errors. Some further provide manually annotated error-type analyses (Zhang et al. (2024); Hao et al. (2025)), but these rely on non-standard category definitions across benchmarks, depend on post-hoc semantic inspection of reasoning traces rather than causal diagnosis, and suffer from annotator variance. In addition, the necessity of multimodal context is often left unverified (Yue et al. (2024b)). Finally, many benchmarks extract problems from publicly available sources such as textbooks, raising risks of data leakage and familiarity bias. To overcome these limitations, our study builds on FormalGeo-7K (Zhang et al. (2023)), which provides formal abstractions for both context and goal in practical mathematical geometry problems, enabling rigorous and fine-grained analysis.

**Training Methods for Multimodal Reasoning.**   Approaches to adapting MLLMs for multimodal reasoning typically fall into three types: multimodal supervised finetuning, where models are trained on paired image–text inputs with ground-truth reasoning traces and answers; multimodal reinforcement learning, where models are optimized with rewards from verifiable outcomes or reasoning-trace feedback; and textual supervised finetuning, where models are tuned on large-scale textual reasoning corpora without multimodal context. These strategies are often applied sequentially or in combination, such as multimodal SFT followed by multimodal RL (e.g., Huang et al. (2025) and Yang et al. (2025)) to improve robustness, or textual SFT followed by MM-RL (e.g., Chen et al. (2025) and  Wei et al. (2025)) to transfer reasoning priors into multimodal domains. In addition, some models adopt direct RL without prior SFT (e.g., Deng et al. (2025), Meng et al. (2025), Wang et al. (2025), and Yao et al. (2025)), demonstrating that reinforcement learning alone can yield competitive reasoning performance. Closed-weight systems (e.g., OpenAI (2025), Comanici et al. (2025), and Anthropic (2025)) also report strong multimodal reasoning ability, although their training data and pipelines remain undisclosed.

## B   COMPARISON WITH EXISTING BENCHMARKS

**Comparison with MathVerse.**   Multimodal mathematical problems are a popular testbed for evaluating multimodal reasoning. Among existing benchmarks (Lu et al. (2024); Wang et al. (2024b); Yue et al. (2024a)), the closest to ours is *MathVerse* (Zhang et al. (2024)), which, like our benchmark, primarily focuses on mathematical geometry problems and partly draws from Geometry3K (Lu et al. (2021)), a subset of FormalGeo-7K (Zhang et al. (2023)). MathVerse also provides multiple input

| **MathVerse_6_TextOnly** | **MathVerse_83_TextOnly** | **MathVerse_98_TextOnly** |
|---|---|---|
| As shown in the figure, AB // CD, and EF intersects AB and CD at points E, F, **angle 1** = 50.0, then the degree of **angle 2** is ()
Choices: A:50° B:120° C:130° D:150° | As shown in the figure, it is known that in circle O, the central angle angle AOB = 100.0, then the angle of circumference angle A**C**B is equal to ().
Choices: A:130° B:120° C:110° D:100° | As shown in the figure, given the angle of circumference angle A = 50.0, then the size of angle O**BC** is ()
Choices: A:50° B:40° C:130° D:80° |

Figure 10: **Example of MathVerse text descriptions.** The textual description fails to fully encode geometric relations, requiring external information to be inferred from the diagram. This incompleteness makes it unsuitable for evaluating pure reasoning capacity.

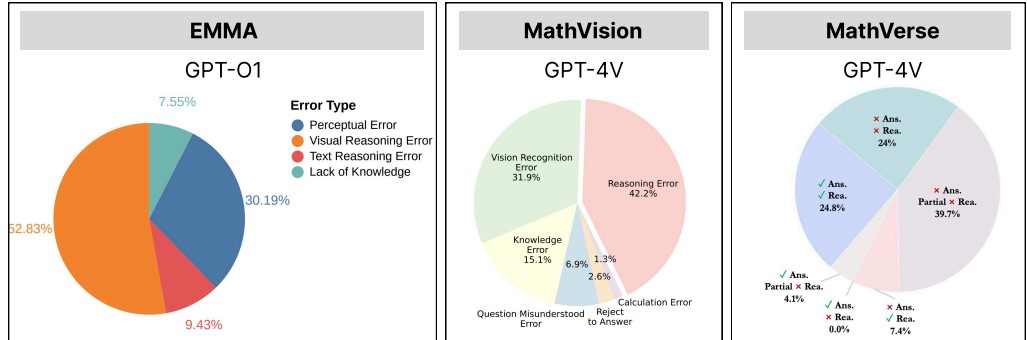

Figure 11: **Inconsistencies in manual error analyses across benchmarks.** Pie charts show variation in error categories across multimodal reasoning datasets, underscoring the lack of standardized criteria. Analyses are typically restricted to a single model. Figures are taken from the respective papers (Hao et al., 2025; Wang et al., 2024b; Zhang et al., 2024).

modalities ranging from text-only to vision-only, enabling skill-specific evaluation of multimodal reasoning models, which is conceptually aligned with our design.

However, MathVerse does not meet the criteria necessary for rigorous capacity isolation: 1) it relies on curated diagrams, making it vulnerable to data leakage or familiarity effects; 2) its text descriptions are incomplete for evaluating pure reasoning, often requiring external information to be inferred from diagrams (see Figure 10); 3) overlap between question and diagram content is not explicitly controlled. Due to these limitations, analyses in this paper cannot be reproduced with the same rigor using prior benchmarks.

**Benefits of automatic error analysis.** Error type analysis offers actionable insights into model weaknesses and guides directions for improvement. However, in multimodal reasoning research, such analyses lack standardization and are almost always performed manually. Consequently, as shown in Figure 11, categories and criteria vary across datasets, making it difficult to generalize findings. Moreover, manual analysis is costly and typically applied to a single model, quickly becoming outdated along with its conclusions. By contrast, MATHLENS's automatic analysis pipeline enables consistent, scalable, and up-to-date error categorization across models and datasets.

## C  DATA GENERATION DETAILS

### C.1  MATHLENS

**Diagram rendering.** Each clause $s_{k,i} \in S_k$ is mapped to a corresponding set of algebraic constraints. For instance, PERPENDICULAR($AB, BC$) is encoded as $((x_a - x_b)(x_c - x_b) + (y_a - y_b)(y_c - y_b) = 0)$. Taken together, the constraints form a nonlinear system defining admissible coordinates for all points. We solve this system using the sequential least-squares programming routine (SLSQP) in `scipy.optimize.minimize` (Virtanen et al. (2020)), which searches for coordinates that minimize the residual of all constraints subject to feasibility bounds. To improve ro-

bustness, initial coordinates are randomly sampled, and optimization is repeated until convergence. Up to ten attempts with different random seeds are allowed, after which the sample is discarded if no feasible solution is found. Valid solutions are rendered with a `matplotlib`-based backend (Hunter (2007)) that draws points, line segments, arcs, and annotations according to the computed geometry. Since automatic optimization may still yield degenerate layouts (e.g., overlapping vertices, occluded labels, or extreme aspect ratios), we apply a post-processing step in which such outputs are manually filtered to preserve clarity and readability.

**Diagram modification.** For each geometry problem, we generate eight diagram variants: (1) the *original* diagram from the source dataset, (2) a *rendered* diagram generated directly from the symbolic representation, and six symbolic modifications: (3) *add_shapes*, which inserts 1–3 random shapes (triangles or quadrilaterals), (4) *add_lines*, which inserts 1–3 random lines between existing points, (5) *flip*, which mirrors the canvas while preserving label orientation, (6) *rotate*, which rotates the canvas while keeping labels upright, (7) *merge*, which concatenates the current diagram with another randomly chosen one and revises labels accordingly, (8) *rename*, which replaces the label set (e.g., A,B,C → X,Y,Z). All auto-rendered figures are manually filtered after generation. If any version is invalid or visually unsuitable (e.g., severe occlusions, degenerate angles), the entire problem is discarded.

### C.2 MATHLENS-GENERAL

MATHLENS-GENERAL is a curated, re-annotated benchmark that extends the scope of MATHLENS to a broader range of images and problem domains. It includes 107 problems, each paired with an average of $\sim 7.96$ visual probe questions. Representative samples are shown in Figure 12 and Figure 13, with the data generation procedure detailed below.

**Sample collection.** Problems are sourced from six established multimodal reasoning datasets: BLINK (Fu et al. (2024)), V* (Wu & Xie (2024)), SpatialEval-Real (Wang et al. (2024a)), MMMU-Pro (Yue et al. (2024b)), EMMA (Hao et al. (2025)), and MathVista (Lu et al. (2024)), with mathematical geometry items excluded using metadata. We retain only multiple-choice problems to maintain consistency. Problems requiring more than two images are discarded, and dual-image inputs are concatenated into single images to ensure compatibility with models that do not support multi-image inputs.

**Data filtering.** Problems that appear multimodal may in fact be solvable from text-only correlations (Yue et al. (2024b)), while others that look complex may reduce to simple pattern matching. To guard against such shortcuts, we use model-based validation to test *multimodality* and *reasoning* requirements. Each problem is evaluated with Gemini-2.5-Flash (Comanici et al. (2025)) under three settings: (1) full input, (2) text-only input without the image, and (3) full input with reasoning disabled. We generate eight responses per setting at temperature 0.6 to capture variability.

A problem passes the *multimodality* check if text-only accuracy does not exceed chance ($1/k$ for $k$-way choice, e.g., 25% for 4 options). It satisfies the *reasoning* check if accuracy with reasoning disabled falls below chance. We enforce *solvability* by discarding problems for which the model fails to answer correctly in all eight full-input attempts. Image-level deduplication is then applied to remove visually similar items.

Finally, human annotators enforce *validity* by independently solving each filtered problem, retaining only those with clearly determinable correct answers. This process reduces the initial pool of $\sim$2,000 problems to 200 high-quality samples that meet all four criteria.

**Manual annotation.** We generate seed annotations with Gemini-2.5-Flash, then manually revise them to remove hallucinations and add missing details needed for solvability. Textual descriptions are structured in a scene-graph format and decomposed into atomic clauses, from which perception probes are automatically derived.

## D  IMPLEMENTATION DETAILS AND RESOURCES

**Models.** Details of the model configurations and corresponding sources are provided in Table 1.

| Model | Type | Source |
|---|---|---|
| **7–9B Open-Weight** | | |
| VL-Rethinker | backbone | `Qwen/Qwen2.5-VL-7B-Instruct` |
| | mm-rl | `TIGER-Lab/VL-Rethinker-7B` |
| ShareVL-R1 | backbone | `Qwen/Qwen2.5-VL-7B-Instruct` |
| | mm-rl | `HuanjinYao/R1-ShareVL-7B` |
| R1-OneVision | backbone | `Qwen/Qwen2.5-VL-7B-Instruct` |
| | mm-sft | `Fancy-MLLM/R1-Onevision-7B` |
| | mm-rl | `Fancy-MLLM/R1-Onevision-7B-RL` |
| Vision-R1 | backbone | `Qwen/Qwen2.5-VL-7B-Instruct` |
| | mm-sft | `Osilly/Vision-R1-CI-7B` |
| | mm-rl | `Osilly/Vision-R1-7B` |
| Revisual-R1 | backbone | `Qwen/Qwen2.5-VL-7B-Instruct` |
| | text-sft | `csfufu/Revisual-R1-Coldstart` |
| | mm-rl | `csfufu/Revisual-R1-final` |
| OVR | backbone | `Qwen/Qwen2.5-VL-7B-Instruct` |
| | text-sft | `Kangheng/OVR-7B-ColdStart` |
| | mm-rl | `Kangheng/OVR-7B-RL` |
| GLM-4.1V | backbone | `zai-org/GLM-4.1V-9B-Base` |
| | mm-rl | `zai-org/GLM-4.1V-9B-Thinking` |
| **72B Open-Weight** | | |
| VL-Rethinker | backbone | `Qwen/Qwen2.5-VL-72B-Instruct` |
| | mm-rl | `TIGER-Lab/VL-Rethinker-72B` |
| **Closed-Weight** | | |
| OpenAI | backbone | **GPT-4O** (`gpt-4o_2024-11-20`) |
| | thinking | **GPT-O3** (`o3_2025-04-16`) |
| Gemini | backbone | `gemini-2.5-flash` (thinking=disabled) |
| | thinking | `gemini-2.5-flash` (thinking=enabled) |
| Claude | backbone | `claude-4-sonnet` (thinking=disabled) |
| | thinking | `claude-4-sonnet` (thinking=enabled) |

Table 1: Model configurations studied in this work.

**Model configuration for Figure 1.** All models are fine-tuned from Qwen-2.5-VL-7B as the backbone MLLM. VL-Rethinker represents the direct RL setting, Vision-R1 serves as the multimodal SFT model, and Revisual-R1 corresponds to the textual SFT model. Vision-R1 and Revisual-R1 further include their respective RL-extended variants.

**Hyperparameters & computation.** We use Eureka ML Insights Framework (Balachandran et al. (2024)) for reproducible evaluation. We run 7–9B parameter models on four NVIDIA A100 80GB GPUs, and 72B models on eight. Generation is accelerated and parallelized with the vLLM (Kwon et al. (2023)) library. Most experiments use greedy decoding (temperature 0.0) for deterministic outputs. For models that otherwise suffer from text degeneration through severe repetition (e.g., Huang et al. (2025)), we apply stochastic decoding with temperature 0.6 and top-$p$ 0.65. The default maximum generation length is 32,768 tokens to accommodate long reasoning chains. For OVR (Wei et al. (2025)), we extend this limit to 48,000 tokens due to frequent truncations at lower cutoffs.

**Visual resources.** Visual icons used in Figure 3 are adapted from `flaticon.com`.

**Large Language Model Usage.** LLMs (ChatGPT, GPT-4/5 class and Claude 4 Sonnet) were employed to refine phrasing, improve clarity, and standardize style in sections of the manuscript, but all scientific ideas, experiments, and analyses were conceived, executed, and validated by the authors. LLMs were also used in a limited capacity to assist with literature discovery (e.g., surfacing

related work for manual screening). All substantive content decisions, experiment design, and result interpretation remain entirely author-driven.

# E   EXPERIMENT

## E.1   PRELIMINARIES

**Weak textual reasoning in multimodal SFT models.**   Prior work (Sun et al. (2025)) highlights that multimodal SFT datasets are considerably easier than textual SFT datasets, which contributes to weaker textual reasoning capacity in trained models. For example, the Vision-R1 dataset (Huang et al. (2025)) averages 821.5 tokens per reasoning trace with a 96.0% pass rate, whereas the text-only DeepMath dataset (He et al. (2025)) averages 8,207.8 tokens with a 75.0% pass rate. This difference suggests that multimodal SFT data require substantially less reasoning effort. Consistent with this, multimodal SFT models trained on such data underperform on standard reasoning benchmarks compared to textual SFT models. Finally, while textual SFT data can be constructed by distilling reasoning traces from large language models (Guo et al. (2025)), no comparably strong multimodal reasoning models with open reasoning traces currently exist, making effective multimodal SFT data generation particularly challenging.

## E.2   FURTHER INSIGHTS

**Insight on visual familiarity effects.**   In Figure 3 (left) of the main paper, we additionally evaluate MATHLENS-E, a variant that uses the same geometry problems but replaces *rendered* diagrams with *existing* diagrams from the original sources. Interestingly, MATHLENS-E correlates strongly with MathVision ($\rho = 0.81$), while showing weaker correlations with MathVista and MathVerse. This difference suggests that models may leverage visual familiarity with diagram styles from textbooks or public tests when tackling MathVision, an advantage that disappears with freshly-rendered diagrams. This underscores that high benchmark accuracy does not necessarily indicate strong multimodal reasoning, as performance may be inflated by visual familiarity effects from training data.

## E.3   MATHLENS-GENERAL

Figure 14 presents the error-type distribution for MATHLENS-GENERAL. Consistent with the main experiment (Section 3.4), most effects of multimodal reasoning finetuning concentrate on perception-related cases (*Perception & Reasoning* or *Perception*). However, these gains are less stable, reflecting that MATHLENS-GENERAL spans broader domains than MATHLENS and often demands out-of-distribution generalization from the multimodal training sets. An exception is textual SFT models, which show substantial reductions in pure *Reasoning* errors. This indicates that, unlike in math geometry tasks, the diverse reasoning skills required for MATHLENS-GENERAL are not well represented in the backbone (Qwen-2.5-VL). Finally, the higher fraction of *Trivial* correct cases arises from MATHLENS-GENERAL 's multiple-choice format, in contrast to the open-ended geometry subset MATHLENS.

## E.4   FULL RESULTS

**Correlation plot details.**   Table 2 reports the full benchmark scores for all models used in the correlation analysis of Figure 5 (left). Results for MathVista, MathVerse, and MathVision are drawn from prior work (Wang et al. (2025)). Consequently, the set of models differs from our main evaluation and includes additional variants such as MM-Eureka (Meng et al. (2025)) and ThinkLite-VL (Deng et al. (2025)). These models were excluded from the main analysis for two reasons: (i) to maintain a balanced number of models across categories, particularly those trained with direct RL, and (ii) to focus on the stronger-performing models on other benchmarks.

**Error type analysis.**   Figure 15 show full error type analysis results for all models tested in this work.

**Downstream performance under diagram modifications.**   Table 3 reports the complete downstream evaluation results on MATHLENS, including all diagram modifications. These values under-

| Model | MathVista | MathVerse | MathVision | **MathLens-E** | **MathLens** |
|---|---|---|---|---|---|
| Qwen2.5-VL-7B | 68.2 | 46.3 | 25.1 | 34.7 | 33.2 |
| R1-Onevision-7B | 64.1 | 46.4 | 29.9 | 35.4 | 29.8 |
| MM-Eureka-Qwen-7B | 73.0 | 50.3 | 26.9 | 34.0 | 31.3 |
| ThinkLite-VL-7B | 74.3 | 52.2 | 29.9 | 34.3 | 32.9 |
| R1-ShareVL-7B | 75.4 | 52.8 | 29.5 | 36.7 | 35.4 |
| VL-Rethinker-7B | 74.9 | 54.2 | 32.3 | 37.6 | 35.7 |
| Qwen2.5-VL-72B | 74.8 | 57.2 | 38.1 | 41.6 | 41.0 |
| VL-Rethinker-72B | 80.4 | 63.5 | 44.9 | 47.6 | 45.0 |

Table 2: Full results of all models used to produce the correlation plot in Figure 5 (left).

| Model | Variant | Text | Raw | Base | Add lines | Add shapes | Flip | Merge | Rename | Rotate | Consistency |
|---|---|---|---|---|---|---|---|---|---|---|---|
| VL-Rethinker-7B | Backbone | 38.9 | 34.7 | 33.2 | 26.1 | 28.1 | 30.6 | 26.9 | 29.4 | 31.4 | 31.0 |
| | MM-RL | 41.1 | 37.6 | 35.7 | 31.4 | 29.4 | 34.7 | 30.6 | 33.5 | 35.5 | 37.3 |
| ShareVL-R1-7B | Backbone | 38.9 | 34.7 | 33.2 | 26.1 | 28.1 | 30.6 | 26.9 | 29.4 | 31.4 | 31.0 |
| | MM-RL | 42.1 | 36.7 | 35.4 | 31.1 | 32.7 | 33.9 | 33.7 | 35.1 | 35.3 | 41.3 |
| R1-OneVision-7B | Backbone | 38.9 | 34.7 | 33.2 | 26.1 | 28.1 | 30.6 | 26.9 | 29.4 | 31.4 | 31.0 |
| | MM-SFT | 36.7 | 35.4 | 29.8 | 21.9 | 26.1 | 29.7 | 25.3 | 27.6 | 29.2 | 24.0 |
| | MM-RL | 35.9 | 33.0 | 32.0 | 25.1 | 28.5 | 34.0 | 23.3 | 30.7 | 32.3 | 28.5 |
| Vision-R1-7B | Backbone | 38.9 | 34.7 | 33.2 | 26.1 | 28.1 | 30.6 | 26.9 | 29.4 | 31.4 | 31.0 |
| | MM-SFT | 35.2 | 34.3 | 26.8 | 19.1 | 21.7 | 27.8 | 20.6 | 23.1 | 28.0 | 21.1 |
| | MM-RL | 48.6 | 44.5 | 36.0 | 33.0 | 31.5 | 36.6 | 30.6 | 34.2 | 37.8 | 39.4 |
| Revisual-R1-7B | Backbone | 38.9 | 34.7 | 33.2 | 26.1 | 28.1 | 30.6 | 26.9 | 29.4 | 31.4 | 31.0 |
| | Text-SFT | 62.3 | 43.8 | 39.2 | 32.9 | 35.1 | 42.7 | 32.8 | 37.4 | 39.8 | 41.0 |
| | MM-RL | 63.8 | 51.7 | 45.8 | 35.3 | 38.0 | 44.4 | 39.1 | 42.2 | 44.7 | 46.9 |
| OVR-7B | Backbone | 38.9 | 34.7 | 33.2 | 26.1 | 28.1 | 30.6 | 26.9 | 29.4 | 31.4 | 31.0 |
| | Text-SFT | 66.2 | 46.7 | 38.7 | 33.8 | 33.7 | 37.9 | 32.5 | 38.1 | 40.7 | 42.9 |
| | MM-RL | 70.4 | 49.8 | 43.7 | 38.4 | 38.2 | 44.9 | 36.2 | 39.7 | 44.9 | 49.9 |
| GLM-4.1V-9B | Backbone | 40.0 | 44.6 | 44.7 | 37.7 | 37.8 | 44.7 | 36.7 | 40.4 | 43.7 | 39.7 |
| | MM-RL | 69.2 | 65.8 | 59.9 | 52.3 | 50.2 | 60.8 | 50.0 | 57.0 | 61.4 | 62.8 |
| VL-Rethinker-72B | Backbone | 52.9 | 41.6 | 41.0 | 36.2 | 35.5 | 42.2 | 37.5 | 38.4 | 41.6 | 40.0 |
| | MM-RL | 56.2 | 47.6 | 45.0 | 38.2 | 39.1 | 45.5 | 38.4 | 44.6 | 44.8 | 43.9 |
| OpenAI 4O / O3 | Backbone | 49.1 | 40.7 | 39.1 | 35.2 | 32.9 | 40.7 | 33.6 | 39.0 | 40.8 | 38.7 |
| | Thinking | 74.5 | 66.8 | 61.4 | 51.9 | 49.9 | 61.7 | 55.8 | 57.8 | 62.9 | 59.0 |
| Gemini-2.5-Flash | Backbone | 79.9 | – | 69.2 | – | – | – | – | – | – | – |
| | Thinking | 82.3 | – | 72.7 | – | – | – | – | – | – | – |
| Claude-4-Sonnet | Backbone | 75.1 | 60.7 | 61.2 | 51.0 | 53.3 | 60.4 | 48.1 | 57.0 | 60.6 | 55.7 |
| | Thinking | 84.1 | 65.6 | 64.4 | 51.0 | 55.0 | 65.6 | 51.5 | 61.0 | 67.0 | 57.0 |

Table 3: Downstream accuracy across diagram modifications with output consistency under these conditions. *Text* indicates performance from textual descriptions instead of diagrams, *Raw* denotes original human-generated diagrams, *Base* the newly rendered diagrams, and the other cases semantic-space modifications followed by rendering.

lie Figure 4, Figure 5 (right), Figure 8, and Figure 7. Note that the error-type analysis in Figure 15 relies on per-sample categorization and cannot be obtained directly from the aggregate scores presented here.

**Perception probe results by question type.** Table 4 reports the full benchmark scores for each perception probe question type, providing the numerical values underlying Figure 9.

# F QUALITATIVE EXAMPLES

**Sample outputs on downstream geometry problems.** We compare the fine-tuned multimodal reasoners with their corresponding backbone MLLMs. Figure 16 and Figure 17 present cases where

| Model | Variant | Tri--angle | Quad--rilateral | Parallel | Perpen--dicular | Collinear | Co--circular | Same len. | Val. len. | Same ∠ | Val. ∠ |
|---|---|---|---|---|---|---|---|---|---|---|---|
| VL-Rethinker-7B | Backbone | 69.1 | 52.3 | 49.0 | 78.2 | 59.5 | 65.6 | 57.4 | 86.6 | 50.0 | 68.1 |
| | MM-RL | 67.6 | 55.2 | 60.6 | 76.6 | 67.4 | 81.7 | 50.0 | 86.9 | 50.0 | 75.8 |
| ShareVL-R1-7B | Backbone | 69.1 | 52.3 | 49.0 | 78.2 | 59.5 | 65.6 | 57.4 | 86.6 | 50.0 | 68.1 |
| | MM-RL | 71.4 | 57.7 | 62.5 | 78.2 | 66.0 | 88.9 | 57.4 | 86.4 | 69.2 | 72.4 |
| R1-OneVision-7B | Backbone | 69.1 | 52.3 | 49.0 | 78.2 | 59.5 | 65.6 | 57.4 | 86.6 | 50.0 | 68.1 |
| | MM-SFT | 67.6 | 48.1 | 56.7 | 67.4 | 53.2 | 60.0 | 48.5 | 76.7 | 46.2 | 63.5 |
| | MM-RL | 71.4 | 54.4 | 59.6 | 72.2 | 61.1 | 59.4 | 55.9 | 79.1 | 61.5 | 66.9 |
| Vision-R1-7B | Backbone | 69.1 | 52.3 | 49.0 | 78.2 | 59.5 | 65.6 | 57.4 | 86.6 | 50.0 | 68.1 |
| | MM-SFT | 65.8 | 49.4 | 43.3 | 73.5 | 64.8 | 60.0 | 45.6 | 78.6 | 61.5 | 73.8 |
| | MM-RL | 70.9 | 56.1 | 49.0 | 76.1 | 65.7 | 77.8 | 57.4 | 88.9 | 61.5 | 76.9 |
| Revisual-R1-7B | Backbone | 69.1 | 52.3 | 49.0 | 78.2 | 59.5 | 65.6 | 57.4 | 86.6 | 50.0 | 68.1 |
| | Text-SFT | 79.1 | 64.0 | 65.4 | 77.5 | 59.8 | 67.8 | 57.4 | 81.7 | 69.2 | 82.3 |
| | MM-RL | 84.3 | 72.0 | 65.4 | 82.8 | 68.0 | 77.2 | 60.3 | 88.1 | 84.6 | 87.8 |
| OVR-7B | Backbone | 69.1 | 52.3 | 49.0 | 78.2 | 59.5 | 65.6 | 57.4 | 86.6 | 50.0 | 68.1 |
| | Text-SFT | 83.5 | 61.5 | 79.8 | 82.2 | 61.0 | 69.4 | 57.4 | 83.2 | 69.2 | 86.0 |
| | MM-RL | 82.2 | 67.8 | 76.9 | 85.9 | 66.9 | 80.6 | 50.0 | 83.0 | 80.8 | 89.0 |
| GLM-4.1V-9B | Backbone | 74.3 | 52.3 | 76.9 | 77.8 | 68.2 | 74.4 | 57.4 | 90.1 | 53.8 | 74.5 |
| | MM-RL | 82.2 | 66.5 | 88.5 | 88.8 | 66.2 | 91.7 | 82.4 | 95.3 | 92.3 | 88.0 |
| VL-Rethinker-72B | Backbone | 66.7 | 49.8 | 76.9 | 80.4 | 66.8 | 94.4 | 67.6 | 94.5 | 69.2 | 87.4 |
| | MM-RL | 83.3 | 57.7 | 79.8 | 84.3 | 65.2 | 96.7 | 64.7 | 93.7 | 80.8 | 86.7 |
| OpenAI 4O / O3 | Backbone | 81.9 | 60.3 | 78.8 | 80.3 | 69.5 | 95.0 | 67.6 | 93.4 | 80.8 | 87.5 |
| | Thinking | 94.0 | 95.4 | 95.2 | 92.8 | 94.1 | 93.9 | 95.6 | 98.9 | 100.0 | 96.7 |
| Gemini-2.5-Flash | Backbone | 92.7 | 90.0 | 97.1 | 92.8 | 89.5 | 96.1 | 95.6 | 97.5 | 88.5 | 92.1 |
| | Thinking | 93.2 | 92.1 | 97.1 | 95.5 | 91.8 | 96.7 | 98.5 | 98.4 | 92.3 | 93.9 |
| Claude-4-Sonnet | Backbone | 88.6 | 75.3 | 79.8 | 85.2 | 82.3 | 95.0 | 77.9 | 98.1 | 84.6 | 92.5 |
| | Thinking | 93.5 | 93.7 | 85.6 | 88.8 | 83.1 | 89.4 | 83.8 | 98.5 | 88.5 | 93.7 |

Table 4: Perception probe accuracy by question types.

fine-tuning corrected an initially wrong answer, while Figure 18 and Figure 19 illustrate cases where the model produced incorrect answers both before and after fine-tuning.

**Sample outputs on perception probes.** Figure 20, Figure 21, and Figure 22 show cases where textual SFT corrected initially wrong answers. Consistent with the quantitative results in Section 3.3, the stronger cognitive patterns induced by textual SFT also promote improved perception.

**Question**

Please directly answer the question and provide the corect option letter, e.g., A, B, C, D.
Question:
Given the following two images, a reference point is annotated on the first image, labeled with REF. You are given multiple red-circled points on the second image, choices of "A, B, C, D" are drawn beside each circle. Select between the choices on the second image and find the corresponding point for the reference point. Which point is corresponding to the reference point?
Choices: A. Point A B. Point B C. Point C D. Point D

**Text Description**

```
{
  ...,
  "parts": [
    {
      "id": "REF",
      "class": "body",
      "part_of": "bird_left",
      "location": "joint of left leg"
    },
    {
      "id": "A",
      "class": "head",
      "part_of": "bird_right",
      "location": "beak"
    },
    {
      "id": "B",
      "class": "leg",
      "part_of": "bird_right",
      "location": "talon of left leg"
    },
    {
      "id": "C",
      "class": "leg",
      "part_of": "bird_right",
      "location": "joint of right leg"
    },
    {
      "id": "D",
      "class": "body",
      "part_of": "bird_right",
      "location": "upper part of left leg"
    }
  ]
}
```

**Perception-Centric QA**

What specific part of the egret is indicated by the point REF?
Choices: A. a leg joint B. a wing joint

Is the point REF located on the left leg of the egret?
Choices: A. right leg B. left leg

Is point A located on the egret's head?
Choices: A. neck B. head

Is point B located on the egret's leg?
Choices: A. leg B. wing

Is point B located on the egret's talon?
Choices: A. talon B. knee

Is point C located on the egret's leg?
Choices: A. body B. leg

Is point C located on a joint of the egret's leg?
Choices: A. leg joint B. leg shaft

Is point C located on the right leg of the egret?
Choices: A. right leg B. left leg

Is point D located on the egret's leg?
Choices: A. body B. leg

Is point D located on the upper part of the egret's leg?
Choices: A. lower part of leg B. upper part of leg

Figure 12: **Data samples from MATHLENS-GENERAL.** We curate problem instance from a prior dataset (Fu et al. (2024)) and annotate the text description and perception-centric question-answers.

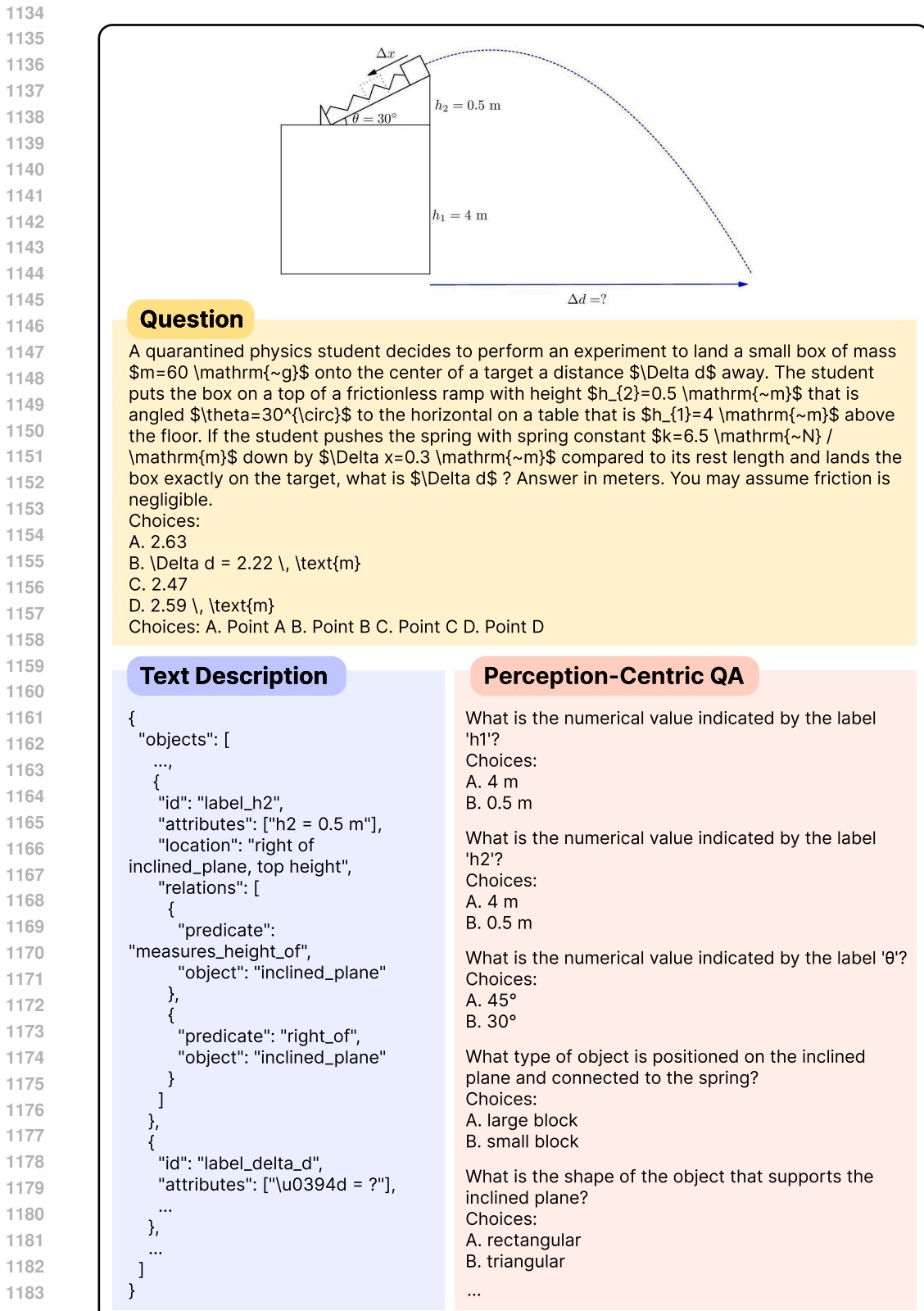

**Question**

A quarantined physics student decides to perform an experiment to land a small box of mass $m=60 \mathrm{~g}$ onto the center of a target a distance $\Delta d$ away. The student puts the box on a top of a frictionless ramp with height $h_{2}=0.5 \mathrm{~m}$ that is angled $\theta=30^{\circ}$ to the horizontal on a table that is $h_{1}=4 \mathrm{~m}$ above the floor. If the student pushes the spring with spring constant $k=6.5 \mathrm{~N} / \mathrm{m}$ down by $\Delta x=0.3 \mathrm{~m}$ compared to its rest length and lands the box exactly on the target, what is $\Delta d$ ? Answer in meters. You may assume friction is negligible.
Choices:
A. 2.63
B. \Delta d = 2.22 \, \text{m}
C. 2.47
D. 2.59 \, \text{m}
Choices: A. Point A B. Point B C. Point C D. Point D

**Text Description**

```
{
  "objects": [
    ...,
    {
      "id": "label_h2",
      "attributes": ["h2 = 0.5 m"],
      "location": "right of
inclined_plane, top height",
      "relations": [
        {
          "predicate":
"measures_height_of",
          "object": "inclined_plane"
        },
        {
          "predicate": "right_of",
          "object": "inclined_plane"
        }
      ]
    },
    {
      "id": "label_delta_d",
      "attributes": ["\u0394d = ?"],
      ...
    },
    ...
  ]
}
```

**Perception-Centric QA**

What is the numerical value indicated by the label 'h1'?
Choices:
A. 4 m
B. 0.5 m

What is the numerical value indicated by the label 'h2'?
Choices:
A. 4 m
B. 0.5 m

What is the numerical value indicated by the label 'θ'?
Choices:
A. 45°
B. 30°

What type of object is positioned on the inclined plane and connected to the spring?
Choices:
A. large block
B. small block

What is the shape of the object that supports the inclined plane?
Choices:
A. rectangular
B. triangular

...

Figure 13: **Data samples from MATHLENS-GENERAL.** We curate problem instance from a prior dataset (Hao et al. (2025)) and annotate the text description and perception-centric question-answers.

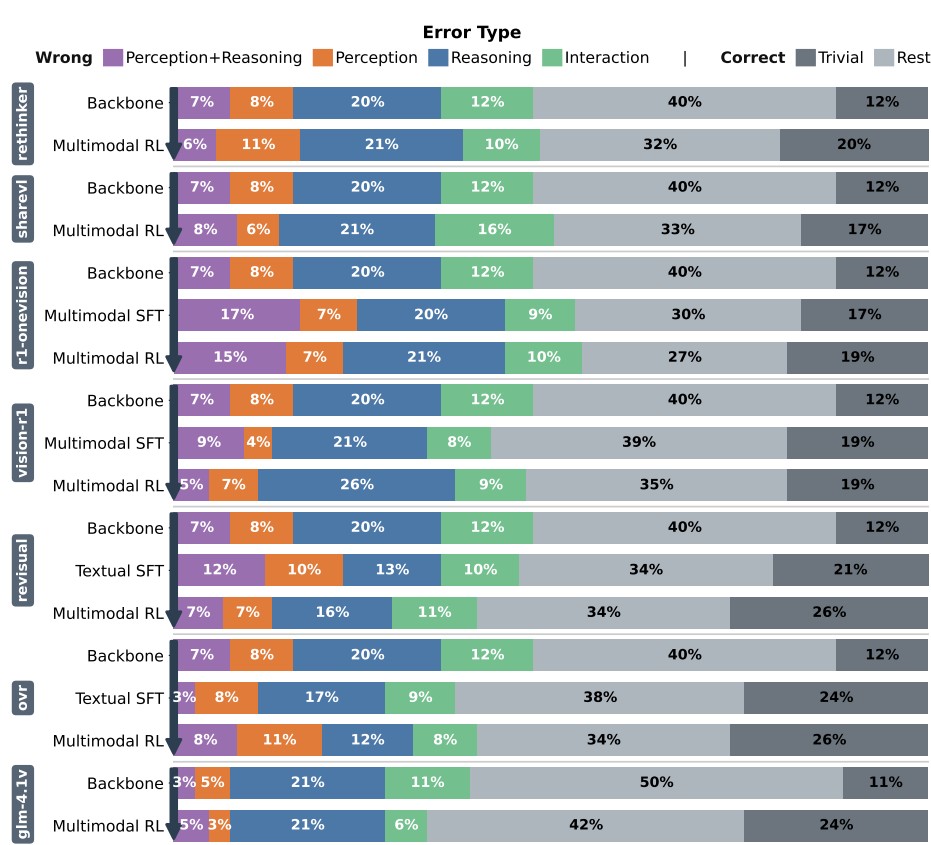

Figure 14: **Error type distribution across models on MATHLENS-GENERAL.** Most shifts are associated with perception-related cases.

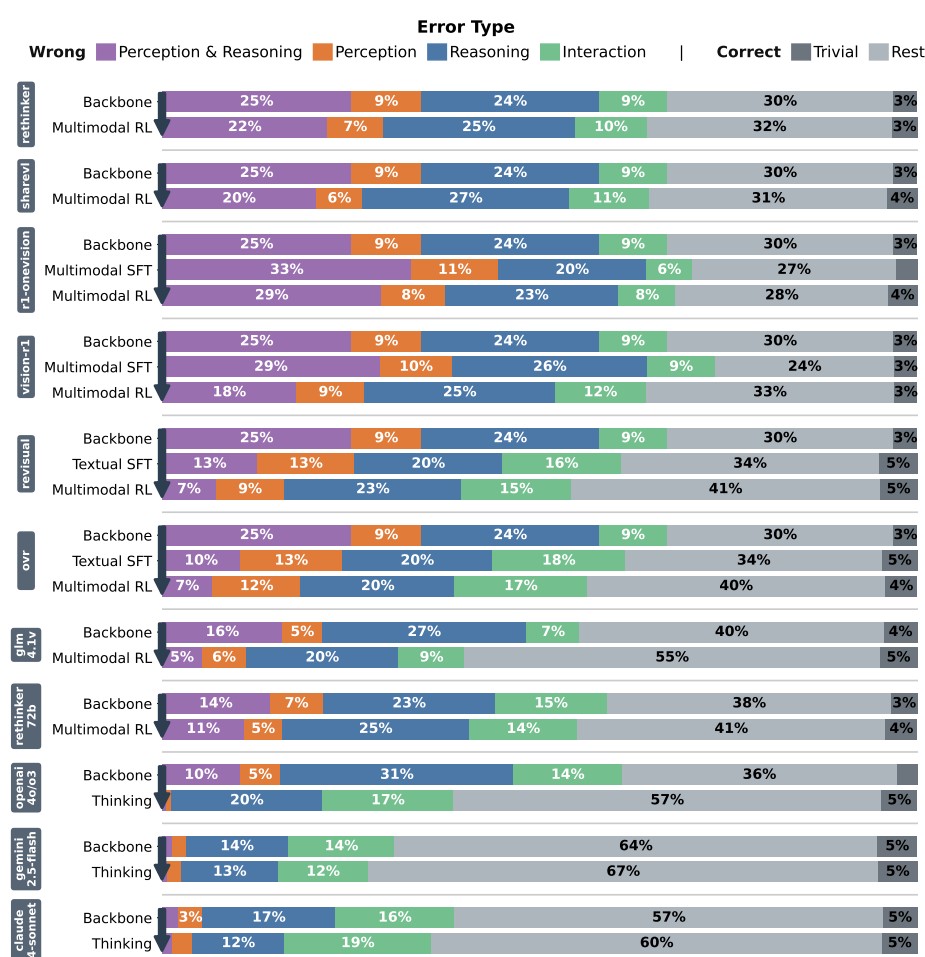

Figure 15: **Distribution of error types per model.** Most gains come from cases failing in both perception and reasoning. Integration errors become more prominent as other categories improve.

**Geometry Problem**

Please first conduct reasoning, and then answer the question and provide the final answer as a symbolic expression or equation involving variables, e.g., 2 \sqrt{x} + 43 or \frac{5 x + 1} {2}, at the end.

**Question: Here is a mathematical geometric problem. As shown in the diagram, find length of line CD. The final answer should be enclosed in \boxed{}.**

**Backbone**: **Qwen-2.5-VL-7B**

To solve for the length of line segment \(CD\), we can use the Pythagorean theorem in the right triangle \(ACD\).
Given:
- **\(AC = 24\)**
- **\(AB = 14\)**
Since \(CD\) is perpendicular to \(AB\), we can split the triangle into two smaller right triangles, \(ACD\) and \(BCD\). However, we only need to focus on triangle \(ACD\) to find \(CD\).
In triangle \(ACD\):
- **\(AC\) is one leg, which is 24.**
- \(AD\) is the other leg, which we need to find.
- **\(CD\) is the hypotenuse.**
Using the Pythagorean theorem in triangle \(ACD\):
\[ AC^2 = AD^2 + CD^2 \]
\[ 24^2 = AD^2 + CD^2 \]
\[ 576 = AD^2 + CD^2 \]
...
Thus, the length of line segment \(CD\) is:
**\[ \boxed{2\sqrt{119}} \]**

**Textual SFT**: **Revisual-R1-7B**

<think>
Okay, let's see. I need to find the length of CD in this triangle. The diagram shows triangle ABC with points A, B, C, and D. The **sides AC and BC are both 24 and 14** respectively.
...
So, triangle ABC, with point D somewhere on AB. **AD is 24, DB is 14**, so AB must be 24 + 14 = 38. Then, CD is perpendicular to both AC and BC? Wait, no. If there's a right angle at C and another at D, maybe CD is perpendicular to AB?
...
Then, AD = 24, DB = 14, and we need to find CD. That seems plausible. Let me confirm.
...
If the problem is structured such that ABC is a right-angled triangle at C, with CD perpendicular to AB, splitting AB into AD = 24 and DB = 14, then CD is indeed sqrt(24*14). But let me verify this formula.
Yes, in a right-angled triangle, the altitude to the hypotenuse is the geometric mean of the two segments. So CD² = AD * DB. Therefore, CD = sqrt(24*14) = sqrt(336). Simplifying sqrt(336): 336 = 16 * 21, so sqrt(16*21) = 4*sqrt(21). Therefore, CD = 4√21. So that would be the answer.
...
</think>
...

**\[\boxed{4\sqrt{21}}\]**

Figure 16: **Model responses for a downstream geometry problem in MATHLENS.** The backbone model fails in perception, leading to an incorrect answer. The textual SFT model initially makes the same mistake but recovers the perceptual detail and produces the correct answer.

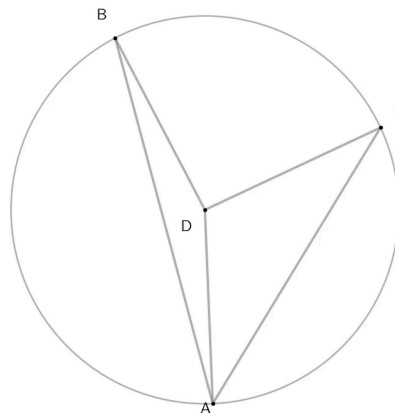

**Geometry Problem**

**Please first conduct reasoning, and then answer the question and provide the final answer as a symbolic expression or equation involving variables, e.g., 2 \sqrt{x} + 43 or \frac{5 x + 1}{2}, at the end.**
**Question: Here is a mathematical geometric problem. As shown in the diagram, measure of arc DBC equals 170. D is centre of circle D. Find measure of angle CAB. The final answer should be enclosed in \boxed{}.**

**Backbone: Qwen-2.5-VL-72B**

To solve for the measure of angle CAB, we need to use properties of circles and angles.

1. **Identify the given information:**
   - The measure of arc DBC is 170 degrees.
   - D is the center of the circle.
2. **Determine the measure of angle BDC:**
   - Since D is the center of the circle, angle BDC is a central angle that intercepts arc BDC.
   - The measure of a central angle is equal to the measure of the intercepted arc.
   - Therefore, angle BDC = 170 degrees.
3. **Use the fact that the sum of angles in triangle BDC is 180 degrees:**
   - In triangle BDC, the sum of the interior angles is 180 degrees.
   - Let angle DBC = x and angle DCB = y.
   - We have: angle BDC + angle DBC + angle DCB = 180 degrees.
   - Substituting the known value: 170 + x + y = 180.
   - Simplifying, we get: x + y = 10.
4. **Determine the measure of angle CAB:**
   - Angle CAB is an inscribed angle that intercepts the same arc BC as the central angle BDC.
   - The measure of an inscribed angle is half the measure of the intercepted arc.
   - The measure of arc BC is 180 - 170 = 10 degrees (since the full circle is 360 degrees and arc DBC is 170 degrees).
   - Therefore, angle CAB = 1/2 * 10 = 5 degrees.

Thus, the measure of angle CAB is **\boxed{5}**.

**Multimodal RL: VL-Rethinker-72B**

To solve the problem, we need to understand the relationship between the arc and the central angle, and then use that to find the inscribed angle.

1. **Identify the central angle:**
   The measure of arc DBC is given as 170 degrees. Since D is the center of the circle, the central angle BDC that subtends this arc is also 170 degrees.

2. **Find the measure of angle BAC:**
   Angle BAC is an inscribed angle that subtends the same arc DBC. The measure of an inscribed angle is half the measure of the central angle that subtends the same arc. Therefore, the measure of angle BAC is:
   \[\text{Measure of angle BAC} = \frac{1}{2} \times \text{Measure of arc BDC} = \frac{1}{2} \times 170 = 85 \text{ degrees}\]

So, the measure of angle CAB is **\boxed{85}**.

Figure 17: **Model responses for a downstream geometry problem in MATHLENS.** The backbone model exhibits a reasoning error, whereas the multimodal RL model provides the correct solution.

**Geometry Problem**

Please first conduct reasoning, and then answer the question and provide the final answer as a symbolic expression or equation involving variables, e.g., 2 \sqrt{x} + 43 or \frac{5 x + 1}{2}, at the end.
**Question: Here is a mathematical geometric problem. As shown in the diagram, find length of line XZ. The final answer should be enclosed in \boxed{}.**

**Backbone: Qwen-2.5-VL-7B**

To solve for the length of line $(XZ)$, **we need to use the Pythagorean theorem since $(XYZ)$ forms a right triangle with $(XY)$ and $(XZ)$ as the legs and $(WY)$ as the hypotenuse.**

Given:
- $(XY = 5x - 7)$
- $(XZ = 3x + 5)$
According to the Pythagorean theorem:
$$ WY^2 = XY^2 + XZ^2 $$
Substitute the given values into the equation:
$$ (5x - 7)^2 = (3x + 5)^2 + (5x - 7)^2 $$
Since both sides of the equation are equal, we can simplify by recognizing that the equation holds true for any $(x)$. However, we need to find the specific value of $(XZ)$.
Let's expand and simplify:
$$ (5x - 7)^2 = (3x + 5)^2 + (5x - 7)^2 $$
$$ 25x^2 - 70x + 49 = 9x^2 + 30x + 25 + 25x^2 - 70x + 49 $$
$$ 25x^2 - 70x + 49 = 25x^2 - 40x + 74 $$
...
Thus, the length of line $(XZ)$ is:
**$$ \boxed{\frac{5}{2}} $$**

**Multimodal RL: VL-Rethinker-7B**

To solve for the length of line $(XZ)$ in the given geometric problem, **we need to use the Pythagorean theorem since $(XYZW)$ forms a right-angled triangle with $(\angle XYZ = 90^\circ)$.**
Given:
- $(XY = 3x + 5)$
- $(YZ = 5x - 7)$
According to the Pythagorean theorem:
$$ XZ^2 = XY^2 + YZ^2 $$
Substitute the given values:
$$ XZ^2 = (3x + 5)^2 + (5x - 7)^2 $$
Expand both squares:
$$ (3x + 5)^2 = 9x^2 + 30x + 25 $$
$$ (5x - 7)^2 = 25x^2 - 70x + 49 $$
...
Let's double check the problem statement and assumptions. If there was an intention to find a specific numerical value, it would likely involve solving for $(x)$ or providing additional information. Since the problem doesn't specify any particular value for $(x)$ and just asks for the length of $(XZ)$, the expression itself is the solution.
Therefore, the length of line $(XZ)$ is:
**$$ \sqrt{34x^2 - 40x + 74} $$**

Figure 18: **Model responses for a downstream geometry problem in MATHLENS.** The multimodal RL model inherits the backbone model's flawed reasoning, leading to perception errors and downstream failure.

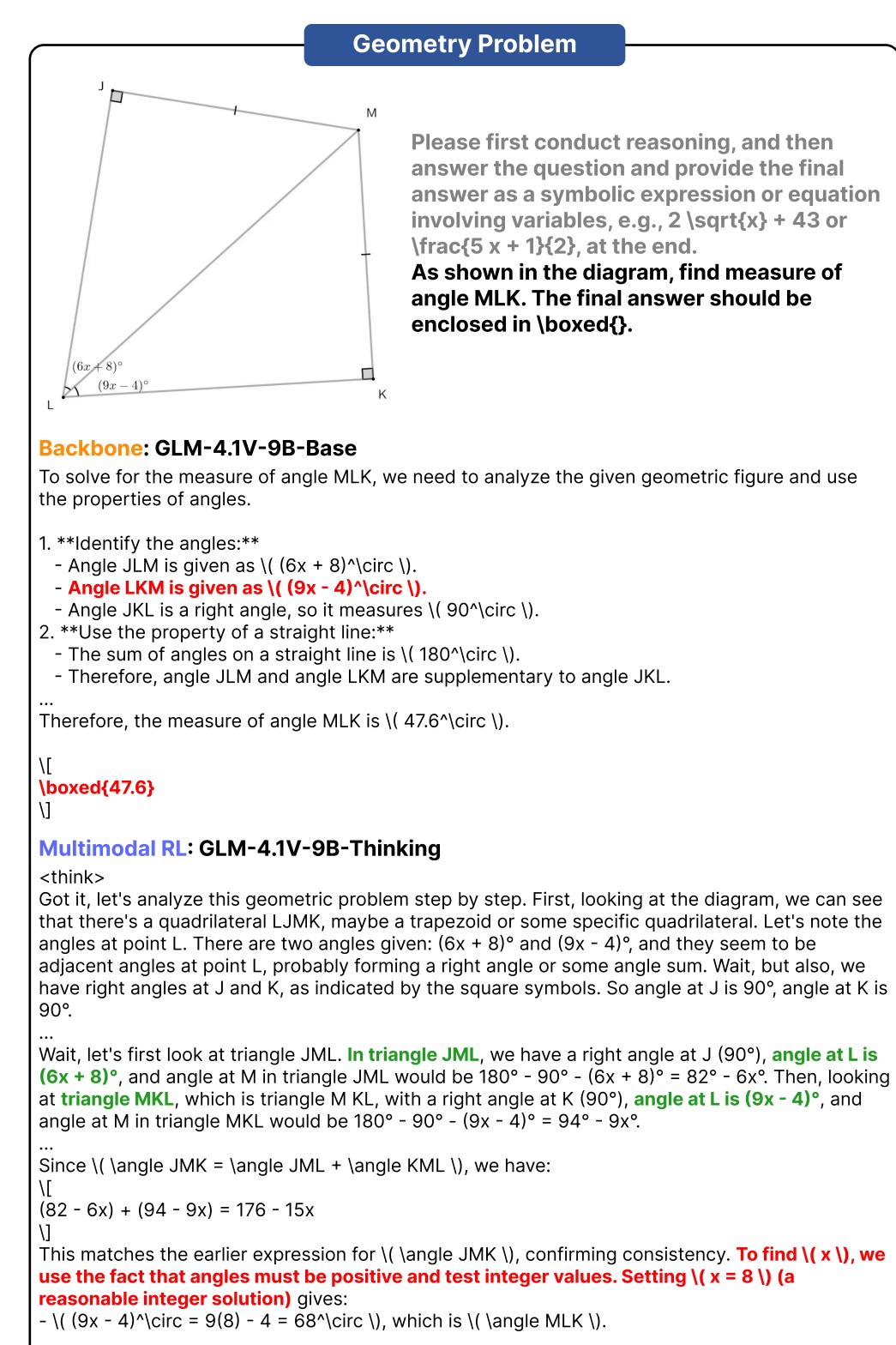

**Geometry Problem**

Please first conduct reasoning, and then answer the question and provide the final answer as a symbolic expression or equation involving variables, e.g., 2 \sqrt{x} + 43 or \frac{5 x + 1}{2}, at the end.
**As shown in the diagram, find measure of angle MLK. The final answer should be enclosed in \boxed{}.**

**Backbone**: GLM-4.1V-9B-Base

To solve for the measure of angle MLK, we need to analyze the given geometric figure and use the properties of angles.

1. **Identify the angles:**
   - Angle JLM is given as \( (6x + 8)^\circ \).
   - **Angle LKM is given as \( (9x - 4)^\circ \).**
   - Angle JKL is a right angle, so it measures \( 90^\circ \).
2. **Use the property of a straight line:**
   - The sum of angles on a straight line is \( 180^\circ \).
   - Therefore, angle JLM and angle LKM are supplementary to angle JKL.
...
Therefore, the measure of angle MLK is \( 47.6^\circ \).

\[
\boxed{47.6}
\]

**Multimodal RL**: GLM-4.1V-9B-Thinking

<think>
Got it, let's analyze this geometric problem step by step. First, looking at the diagram, we can see that there's a quadrilateral LJMK, maybe a trapezoid or some specific quadrilateral. Let's note the angles at point L. There are two angles given: (6x + 8)° and (9x - 4)°, and they seem to be adjacent angles at point L, probably forming a right angle or some angle sum. Wait, but also, we have right angles at J and K, as indicated by the square symbols. So angle at J is 90°, angle at K is 90°.
...
Wait, let's first look at triangle JML. **In triangle JML**, we have a right angle at J (90°), **angle at L is (6x + 8)°**, and angle at M in triangle JML would be 180° - 90° - (6x + 8)° = 82° - 6x°. Then, looking at **triangle MKL**, which is triangle M KL, with a right angle at K (90°), **angle at L is (9x - 4)°**, and angle at M in triangle MKL would be 180° - 90° - (9x - 4)° = 94° - 9x°.
...
Since \( \angle JMK = \angle JML + \angle KML \), we have:
\[
(82 - 6x) + (94 - 9x) = 176 - 15x
\]
This matches the earlier expression for \( \angle JMK \), confirming consistency. **To find \( x \), we use the fact that angles must be positive and test integer values. Setting \( x = 8 \) (a reasonable integer solution)** gives:
- \( (9x - 4)^\circ = 9(8) - 4 = 68^\circ \), which is \( \angle MLK \).
...
The answer is \(**<|begin_of_box|>68<|end_of_box|>**\).

Figure 19: **Model responses for a downstream geometry problem in MATHLENS.** The multimodal RL model corrects the perceptual error of the backbone model but ultimately fails to produce the correct answer due to a reasoning error.

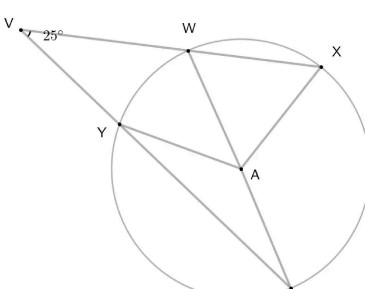

**Perception-Centric QA**

Please first conduct reasoning, and then answer the question and provide the correct option letter between A or B at the end. Use only explicit information from the image without making inferences.
**Question: Which sequence of points is collinear?**
**Choices:**
**A. VWZ**
**B. VYZ**

**Backbone**: Qwen-2.5-VL-7B

To determine which sequence of points is collinear, we need to check if the points lie on the same straight line.
From the diagram:
- Points V, W, and Z are connected by lines that form a triangle with point A inside it.
- Point Y is also part of the diagram but does not seem to be directly connected to V, W, or Z in a way that suggests collinearity with them.
Since there is no direct indication that points V, W, and Z are collinear (they form a triangle), and point Y is not connected to V, W, or Z in a way that suggests collinearity, the correct sequence of points that appears to be collinear is:

\boxed{A}

**Textual SFT**: Revisual-R1-7B

<think>
Okay, let's see. The question is asking which sequence of points is collinear. The options are A. VWZ and B. VYZ. The image provided is a circle with several points labeled: V, W, X, Y, Z, and A. There's also a 25-degree angle mentioned at point V.
First, I need to recall what collinear means. Collinear points are points that lie on the same straight line. So, I need to check if the points in each option lie on a single straight line in the given diagram.
...
But since the problem mentions a 25-degree angle at V, and the options are VWZ and VYZ, perhaps the correct answer is B. VYZ. **Because if the angle at V is between VW and VY, and Z is on the line VY**, then VYZ are collinear. Alternatively, if the angle is between VW and VZ, then VWZ are collinear.
...
</think>
To determine which sequence of points is collinear, we analyze the given information: points V, W, X, Y, Z, and A lie on a circle, and there is a 25° angle at point V. Collinear points must lie on the same straight line.
1. **Option A (VWZ):** If V, W, Z are collinear, the line VWZ would pass through W and Z on the circle. However, the 25° angle at V suggests a relationship between two lines from V intersecting the circle. If the angle is between VW and VY, Z would need to lie on VW to make VWZ collinear, but this is not explicitly stated.
2. **Option B (VYZ):** If V, Y, Z are collinear, the line VYZ would pass through Y and Z on the circle. Given the 25° angle at V, it is plausible that this angle is formed by secants VW and VY intersecting the circle. If Z lies on the line VY, then VYZ would be collinear. This configuration aligns with typical geometric setups where angles at external points relate to secants intersecting the circle.
Since the problem specifies using only explicit information (the 25° angle at V and the circle containing V, W, X, Y, Z, A), the most logical conclusion is that the 25° angle involves secants VW and VY, with Z positioned on the line VY. Thus, **VYZ (Option B)** is collinear.

\boxed{B}

Figure 20: **Model responses for a perception probe in MATHLENS.** The textual SFT model corrects the perceptual error present in the backbone model.

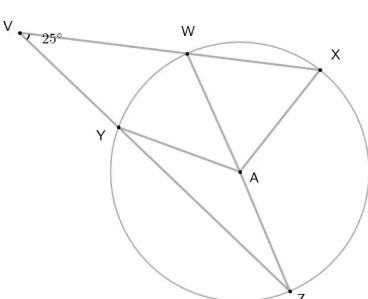

**Perception-Centric QA**

**Please first conduct reasoning, and then answer the question and provide the correct option letter between A or B at the end. Use only explicit information from the image without making inferences.**
**Question: Which angle has a measure of 25 degrees?**
**Choices:**
**A. YVW**
**B. ZVY**

**Backbone: Qwen-2.5-VL-7B**

To determine which angle has a measure of 25 degrees, we need to look at the given information in the image.
1. The image shows a circle with points V, W, X, Y, and Z.
2. There is an angle labeled as 25 degrees, and it appears to be an exterior angle formed by the line segment VY and the tangent at point V.
Given this information, the angle that measures 25 degrees is the one formed by the line segment VY and the tangent at point V, which corresponds to angle ZVY.
Therefore, the correct answer is:

\boxed{B}

**Textual SFT: Revisual-R1-7B**

<think>
Okay, let's see. The question is asking which angle has a measure of 25 degrees, and the options are A. YVW and B. ZVY. The image provided is a circle with several points labeled: V, W, X, Y, Z, and A. There's a 25-degree angle mentioned at point V.
First, I need to recall how angles in a circle work. If there's a 25-degree angle at point V, that might be an inscribed angle or maybe a central angle. But in the diagram, the angle at V is between points Y and W. Wait, the angle is labeled as 25 degrees at V, so the angle at vertex V between points Y and W. So angle YVW is 25 degrees. That would be option A. But wait, the choices are A. YVW and B. ZVY. So if the angle at V is 25 degrees, then A is the answer. But maybe I need to check if there's another angle related to that.
...
Looking at the image, the points are connected as follows: V is connected to Y and W (since angle YVW is 25 degrees). Then there's a circle with points Y, W, X, Z, etc. Maybe there are chords connecting these points. For example, Y to W, W to X, X to Z, Z to Y, etc. But the key is that the angle at V is 25 degrees, which is angle YVW. So that's angle A. Then angle ZVY would be at point V between Z and Y. But unless there's a line from V to Z, that angle isn't present. The problem states to use only explicit information. So if there's no line from V to Z, then angle ZVY doesn't exist. Therefore, the 25-degree angle is angle YVW, which is option A.
...
Wait, but maybe the 25 degrees is part of another angle. For example, if there's a triangle or another angle formed by intersecting chords. Let me think. If two chords intersect at a point, the measure of the angle is half the sum or difference of the intercepted arcs. But in this case, the angle at V is given as 25 degrees. If that's an inscribed angle, then the intercepted arc would be 50 degrees. But the problem states to use only explicit information. The explicit information is that there's a 25-degree angle at V between Y and W. So unless there's another angle at V that's 25 degrees, which would be ZVY, but there's no line from V to Z shown.
</think>
The angle at point V between points Y and W is explicitly labeled as 25 degrees in the diagram. This corresponds to angle YVW. There is no explicit mention or indication of a line from V to Z, so angle ZVY cannot be confirmed.

\boxed{A}

Figure 21: **Model responses for a perception probe in MATHLENS.** The textual SFT model corrects the perceptual error present in the backbone model.

1620
1621
1622
1623
1624
1625
1626
1627
1628
1629
1630
1631
1632
1633
1634
1635
1636
1637
1638
1639
1640
1641
1642
1643
1644
1645
1646
1647
1648
1649
1650
1651
1652
1653
1654
1655
1656
1657
1658
1659
1660
1661
1662
1663
1664
1665
1666
1667
1668
1669
1670
1671
1672
1673

## Perception-Centric QA

Please first conduct reasoning, and then answer the question and provide the correct option letter between A or B at the end. Use only explicit information from the image without making inferences.
**Question: Which angle has a measure of 35 degrees?**
**Choices:**
**A. CBA**
**B. BCA**

**Backbone: Qwen-2.5-VL-7B**

To determine which angle has a measure of 35 degrees, we need to identify the angle labeled as 35 degrees in the diagram.
From the image, we can see that the angle labeled as **35 degrees is located at vertex B** and is part of the triangle ABC.
Therefore, the angle with a measure of 35 degrees is BCA.

The correct answer is **\boxed{B}**.

**Textual SFT: Revisual-R1-7B**

<think>
Okay, let's see. I need to figure out which angle in the given triangle is 35 degrees. The triangle has points A, B, and C. The given information is that **angle at B is 35 degrees**, angle at C is 52 degrees, and there's an angle at A labeled as x degrees. The choices are A. CBA and B. BCA. Wait, let me make sure I understand the notation here.
First, angle **CBA would be the angle at point B, right? Because the middle letter is the vertex**. So CBA is angle at B. Similarly, BCA is the angle at C. So the question is asking which of these two angles (at B or at C) is 35 degrees. But wait, the problem states that angle B is 35 degrees. So that would be angle CBA, which is option A. But wait, let me check again.
...
Wait, the problem says "angle B" is 35°, so angle at B is 35°, which is angle CBA. So that's option A. But maybe I'm missing something. Let me check the image description again. The image shows a triangle with points A, B, C. The angle at B is 35°, angle at C is 52°, and angle at A is x°. So the angles are: B = 35°, C = 52°, A = 93°. Therefore, the angle with 35° is angle CBA (at B), which is option A. So the answer should be A. But wait, the problem might have a different labeling. Let me make sure.
Wait, sometimes in geometry diagrams, the labels can be tricky. Let me think. If the triangle is labeled with vertices A, B, C, then angle CBA is the angle at B between points C and A. Similarly, angle BCA is the angle at C between points B and A. So in the given problem, angle at B is 35°, which is angle CBA (option A). The angle at C is 52°, which is angle BCA (option B). So the 35° angle is angle CBA, so the answer is A. Therefore, the correct option is A.
</think>
The triangle has angles at points B, C, and A. The given angles are 35° at B and 52° at C. To find the measure of angle A, we use the triangle angle sum property:
\[
35^\circ + 52^\circ + x = 180^\circ \implies x = 180^\circ - 35^\circ - 52^\circ = 93^\circ.
\]
However, the question asks which angle is **35°**, which is explicitly labeled at point B. The angle at B is denoted as **CBA** (with vertex at B), corresponding to option **A**.

**\boxed{A}**

Figure 22: **Model responses for a perception probe in MATHLENS.** The textual SFT model corrects the perceptual error present in the backbone model.

