# OpenReview forum: "What MLLMs Learn about When they Learn about Multimodal Reasoning: Perception, Reasoning, or their Integration?"
_ICLR.cc/2026/Conference — Submitted to ICLR 2026_

### Official Review · Reviewer_sN3K · 2025-10-15

**Soundness:** 2
**Presentation:** 2
**Contribution:** 2
**Rating:** 6
**Confidence:** 3

**Summary:**

They introduce MATHLENS, a benchmark designed to disentangle the subskills of multimodal reasoning while preserving the complexity of textbookstyle geometry problems. The benchmark separates performance into three components: Perception: extracting information from raw inputs, Reasoning: operating on available information, and Integration: selecting relevant perceptual evidence and applying it within reasoning.

**Strengths:**

1. The paper is clearly written with demonstrative figures.
2. The idea of separating the reasoning process of MLLMs is interesting.
3. The analysis of the result of different methods is detailed and shines light on the inner workings of MLLM reasoning.

**Weaknesses:**

1. The perception probes test for a finite set of atomic facts. A model might correctly identify all the probed facts but fail to perceive another crucial, un-probed visual detail. This perceptual failure would be misclassified as an integration failure.
2. The primary benchmark, MATHLENS, is composed entirely of geometry problems. It cannot catch the full scope of MLLM reasoning. The add-on MATHLENS-GENERAL cannot maintain the same rigor.

**Questions:**

See weaknesses.

---

> ### Author Response · Authors · 2025-11-17
>
> We thank the reviewer for acknowledging the clarity of our presentation, the interest of our reasoning-separation framework, and the value of our analyses in illuminating the inner workings of multimodal reasoning. We address the concerns below:
>
> ---
>
> ## **Weakness 1. Integrity of the perception probes**
>
> Our perception probes are designed to cover *all* visual details required to solve each problem. This is ensured by the construction pipeline:
>
> - Each geometry problem is sourced from FormalGeo-7K, which provides a symbolic representation of the diagram as a set of formal clauses (e.g., `Perpendicular(AB, BC)`).
> - These clause sets are *complete*: they are manually curated and formally verified in FormalGeo to be sufficient to solve the problem.
> - We convert each clause into a visual question, which becomes a perception probe (e.g., *“Are AB and BC perpendicular?”*).
> - Because every clause is translated, the perception probes collectively capture **all** perceptual information relevant to the solution.
> - As a result, the type of “misclassification” scenario raised by the reviewer cannot arise in our benchmark; eliminating such ambiguity is one of its key strengths.
>
> We will clarify this construction more explicitly in the revision.
>
> ---
>
> ## **Weakness 2. Focus on geometry domain**
>
> First, **geometry is a canonical testbed for multimodal reasoning**. Widely used benchmarks [1] in this space mostly revolve around mathematics [2,3,4], and one consists exclusively of geometry problems [2]. Mathematics is a foundational domain for textual reasoning [5], and geometric questions uniquely allow simultaneous evaluation of perception, symbolic reasoning, and cross-modal integration. This makes geometry a natural and widely adopted setting for studying multimodal reasoning behavior.
>
> While we agree that incorporating additional domains would broaden the benchmark, focusing on geometry remains useful because it provides a well-structured and widely studied environment for analyzing perception–reasoning interactions, while not intended to cover all possible task types.
>
> Second, **MathLens-General is nearly as rigorous**. We performed extensive manual annotation to ensure that the criteria established in the geometry subset transfer faithfully to the broader benchmark as detailed in Appendix C.2. However, unlike geometry, where structure enables systematic annotation, general visual reasoning tasks rely more heavily on manual labeling, which limits scalability. Nonetheless, we ensured that the same principles were applied as consistently as possible.
>
> ---
>
> ### **References**
>
> [1] OpenCompass Leaderboard (https://huggingface.co/spaces/opencompass/Open_LMM_Reasoning_Leaderboard)
>
> [2] MathVerse, ECCV 2024 (https://arxiv.org/pdf/2403.14624)
>
> [3] MathVision, NeurIPS D&B 2024 (https://arxiv.org/pdf/2402.14804)
>
> [4] MathVista, ICLR 2024 (https://arxiv.org/pdf/2310.02255)
>
> [5] AIME 2025 (https://huggingface.co/datasets/MathArena/aime_2025)

---

> ### Comment · Reviewer_sN3K · 2025-11-19
> **Response to Official Comment by Authors**
>
> Thank you for the clarification. I will maintain my score and vote in favor of accepting this paper.

---

### Official Review · Reviewer_atqw · 2025-10-25

**Soundness:** 2
**Presentation:** 3
**Contribution:** 2
**Rating:** 4
**Confidence:** 4

**Summary:**

This paper addresses the problem that aggregate accuracy metrics are insufficient for evaluating and understanding the progress of Multimodal Large Language Models (MLLMs) on complex reasoning tasks. By comparing a model's performance across these tests, the authors can decompose errors into failures of perception, reasoning, or Integration (defined as failing the joint task despite succeeding on perception and reasoning individually). The methodology also involves generating semantic variations of diagrams to test robustness.

**Strengths:**

- Error analysis reveals RL shifts errors to integration, providing evidence-based guidance for future training, unlike aggregate benchmarks.
- Use of symbolic states ensures equivalence across modalities, supporting valid isolation of subskills with high diagnostic value.
- The evaluation of robustness using semantic-level diagram modifications, rather than just pixel-level noise.

**Weaknesses:**

- The primary analysis and all major findings are derived exclusively from the geometry domain. While this allows for tight control, it leaves the generalizability of the conclusions.
- The accuracy of the error decomposition hinges on the assumption that the perception probes are exhaustive.
- The paper presents many empirical comparisons but does not report confidence intervals or use statistical tests to confirm the significance of the observed differences.

**Questions:**

- Regarding the error decomposition, how did you ensure the set of perception probes for each problem was comprehensive? Is it possible that some errors classified as "integration" failures are in fact subtle perceptual failures not captured by the probes?
- What compute was used for evaluations, and how might nondeterminism affect API models?

---

> ### Author Response · Authors · 2025-11-17
>
> We thank the reviewer for acknowledging the clarity of our analysis and the diagnostic value of our framework, including the insights from error patterns under RL, the use of symbolic states for modality-equivalent evaluation, and the robustness tests based on semantic rather than pixel-level perturbations. We address the concerns point by point below.
>
> ---
>
> ## **Weakness 1. Focus on geometry domain**
>
> **Geometry is a canonical testbed for multimodal reasoning**. Widely used benchmarks in this space [1–4] predominantly focus on mathematics, and one evaluates geometry exclusively [2]. Mathematics is a foundational setting for textual reasoning [5], and geometric problems uniquely allow simultaneous evaluation of perception, symbolic reasoning, and cross-modal integration. This makes geometry a natural and well-established environment for controlled analysis of multimodal reasoning behaviors.
>
> While incorporating additional domains would broaden coverage, focusing on geometry remains useful because it provides a structured and widely studied setting for isolating perception–reasoning interactions. It is not intended to cover all possible task types, but it aligns with how the field typically probes multimodal reasoning.
>
> In addition, **we also provide a broader benchmark, MathLens-General** in Section 2.3 and Appendix C.2 and E.3. We performed extensive manual annotation to transfer the same criteria to this general domain set. Unlike geometry, where structure enables systematic annotation, general visual reasoning tasks rely on more manual labeling, which limits scalability. Nevertheless, we ensured the same principles were applied consistently.
>
> ---
>
> ## **Weakness 2 & Question 1. Completeness of perception probes**
>
> Our perception probes are *exhaustive by construction*. The pipeline ensures this:
>
> - Each geometry problem originates from FormalGeo-7K, which includes a symbolic clause set describing the diagram (e.g., `Perpendicular(AB, BC)`).
> - These clause sets are manually curated and formally verified in FormalGeo to be sufficient to derive the correct solution.
> - Every clause is translated into a corresponding visual question, forming the perception probe set (e.g., *“Are AB and BC perpendicular?”*).
> - Therefore, the probes collectively encode **all** perceptual information required to solve the problem.
> - As each clause is included, errors attributed to “integration” cannot be due to missing perceptual information.
>
> We will clarify this more explicitly in the revision.
>
> ---
>
> ## **Weakness 3. Statistical significance and decoding determinism**
>
> As described in Section 3 and Appendix D, nearly all models use greedy decoding (temperature = 0), producing deterministic outputs. In such settings, confidence intervals and significance tests are not meaningful. Two multimodal-SFT models required sampling due to severe repetition instability; the rest behave deterministically under our setup.
>
> We discuss nondeterminism in API models below.
>
> ---
>
> ## **Question 2. Compute details and nondeterminism in closed APIs**
>
> **Compute.**
>
> For open-source models, we used two nodes with 8×A100 (80GB) GPUs each. We used tensor_parallel_size=4 for smaller (\~8B) models and 8 for larger (\~72B) models and conducted multiple experiments in a parallel manner.
>
> **Nondeterminism in closed APIs.**
>
> Closed API models occasionally return different outputs even with temperature = 0. The source of this nondeterminism is unclear due to their black-box nature; kernel-level nondeterminism has been reported as a possible cause [6]. This phenomenon is not handled in existing literature, and repeatedly querying APIs would incur prohibitive costs. We therefore do not attempt to correct or control for this nondeterminism.
>
> ---
>
> ### **References**
>
> [1] OpenCompass Leaderboard (https://huggingface.co/spaces/opencompass/Open_LMM_Reasoning_Leaderboard)
>
> [2] MathVerse, ECCV 2024 (https://arxiv.org/pdf/2403.14624)
>
> [3] MathVision, NeurIPS D&B 2024 (https://arxiv.org/pdf/2402.14804)
>
> [4] MathVista, ICLR 2024 (https://arxiv.org/pdf/2310.02255)
>
> [5] AIME 2025 (https://huggingface.co/datasets/MathArena/aime_2025)
>
> [6]  “Defeating Nondeterminism in LLM Inference,” (https://thinkingmachines.ai/blog/defeating-nondeterminism-in-llm-inference/)

---

> > ### Comment · Reviewer_atqw · 2025-11-19
> > **My Concerns Remain Unaddressed**
> >
> > Thank you for your response. However, my concerns remain unaddressed. This paper makes strong empirical claims but lacks necessary methodological rigor.
> >
> > **On Statistical Significance.** Your claim that deterministic decoding eliminates the need for significance testing misunderstands benchmark methodology. You are making comparative claims across models ("RL improves perception by X%", "textual SFT reduces robustness") that require statistical validation to demonstrate differences are meaningful rather than artifacts of evaluation variance or error propagation across your multi-stage decomposition. Figure 8 shows substantial variance across diagram modifications, necessitating significance testing. Benchmark papers cannot avoid statistical validation of comparative claims.
> >
> > **On Evaluation Procedures.** You claim to disentangle perception, reasoning, and integration, but never specify how outputs are evaluated. How are perception probe answers judged, multiple choice matching, free-form parsing, LLM-as-judge? What thresholds determine success for each component? How do you handle partial success when categorizing integration failures? Without these details, your benchmark cannot be reproduced. This is the core evaluation methodology left unspecified.
> >
> > **On Human Validation.** You need human annotators to verify your automated evaluation correctly judges model outputs. A rigorous benchmark requires demonstrating high agreement between human judgments and automated scoring across representative samples, with reported correlation or agreement metrics. You have provided no such validation, leaving your evaluation pipeline unverified.
> >
> > **On Domain Limitation.** MathLens-General does not resolve domain concerns. With only 107 problems using looser controls, it cannot support generalizable conclusions about 13+ models.

---

> > > ### Author Response · Authors · 2025-11-22
> > >
> > > We appreciate the reviewer’s detailed follow-up and understand the concern regarding methodological rigor. We will release all reasoning traces and evaluation scripts for reproducibility. We address each point below.
> > >
> > > ---
> > >
> > > ## **1. On statistical significance**
> > >
> > > We agree that comparative claims should not depend on unstable measurements. We will report any remaining variance (including statistical tests where appropriate), and we welcome clarification if there are additional sources we may have overlooked.
> > >
> > > Our evaluation was designed to remove the standard sources of stochastic variance that significance testing typically addresses, and the diagram perturbations in Figure 8 are not instances of such variance but controlled robustness conditions.
> > >
> > > ### **Controlled sources of stochastic variance**
> > >
> > > Benchmark scores commonly vary due to several sources of noise:
> > >
> > > - Sampling variance: different evaluation subsets across runs.
> > > - Decoding variance: stochastic sampling or nucleus sampling.
> > > - Randomized preprocessing: shuffling, augmentations, or random transformations.
> > >
> > > We explicitly controlled these factors:
> > >
> > > - All models are evaluated on the full fixed benchmark without subsampling.
> > > - Decoding is fully deterministic.
> > > - No randomized perturbations or stochastic augmentations are applied.
> > >
> > > With these controls, re-running the evaluation produces identical scores, so the run-to-run variance that significance tests usually target does not arise.
> > >
> > > ### **On diagram perturbations**
> > >
> > > The diagram modifications in Figure 8 (occlusion, rotation, altered labels, added distractors) are not random draws from a shared distribution. They are **predefined, systematic shifts** introduced to probe specific robustness dimensions:
> > >
> > > - Each perturbation alters perceptual cues and task difficulty in a directed, non-random way.
> > > - They form **distinct evaluation conditions**, not IID resamples.
> > > - Differences across them therefore reflect **intended robustness effects**, not statistical noise.
> > >
> > > Thus, Figure 8 captures performance under deliberately shifted diagram conditions, not variability from randomness in the evaluation pipeline.
> > >
> > > ### **On error propagation**
> > >
> > > The decomposition is not a multi-stage pipeline: perception, reasoning, and downstream scores are obtained through independent deterministic evaluations, and “integration” is an algebraic residual rather than the output of a downstream stage. Thus, there is no stage-to-stage error propagation or stochastic coupling.
> > >
> > > ---
> > >
> > > ## **2. On Evaluation Procedures**
> > >
> > > We will clarify the procedures in the revised draft and release the complete evaluation script to ensure full reproducibility. The scoring pipeline is fully deterministic and does not rely on any LLM-as-judge components.
> > >
> > > **Perception probes.**
> > >
> > > Perception probes are binary classification tasks with hard negatives. Predictions are evaluated by exact match against the ground-truth index. No thresholds, similarity metrics, or heuristic scoring methods are used.
> > >
> > > **Reasoning answers.**
> > >
> > > Reasoning probes and final answers are evaluated using an established deterministic numeric or symbolic extractor [1] followed by an exact match.
> > >
> > > **Component success and partial credit.**
> > >
> > > The perception component is considered correct only if all associated probes for that component are correct. We do not assign partial credit, as fractional scoring may inadequately represent the final correctness.

---

> ### Author Response · Authors · 2025-11-22
>
> ## **3. On Human Validation**
>
> We agree that evaluation pipelines should be transparent and verifiable. However, human annotation is not a gold label in this setting. For LLMs, chain-of-thought traces are not guaranteed to reflect the causal computation that produced the final answer [2], so humans cannot infer the true source of an error from the output alone.
>
> That said, we additionally performed a small-scale human evaluation on 100 randomly sampled outputs per GPT-4O model in MathLens, following the protocol and data scale used in prior work [3,4]. Due to the high cost of manually inspecting reasoning trajectories, this evaluation cannot be easily scaled to achieve statistical significance. Human annotators were instructed to determine the error type from the model’s reasoning trace on the downstream task.
>
> The results show strong alignment between human evaluation and our automatic evaluator for perception and reasoning errors. All instances that auto-eval labeled as *perception-only* errors (5/5) were also labeled as perception errors by humans. Likewise, all cases that humans labeled as *reasoning-only* errors (26/26) were also flagged as reasoning errors by auto-eval. For outputs that auto-eval labeled as both perception and reasoning errors (10/10), humans predominantly attributed the failure to perception (9/10), which is consistent with the fact that downstream reasoning becomes invalid once an early perceptual mistake is made.
>
> In contrast, interaction-type errors are by definition impossible to recover from post-hoc inspection of the reasoning traces alone. Among the 14 interaction errors flagged by auto-eval, human annotators reclassified 5 as perception errors and 9 as reasoning errors, reflecting the inherent ambiguity of attributing failures to the interaction category without access to intermediate model internals.
>
> We will report the human evaluation results in a dedicated chart in the revised manuscript.
>
> ---
>
> ## **4. On domain limitation**
>
> We would like to clarify that the size of MathLens-General reflects our choice to prioritize quality over quantity. Each sample was filtered extensively to require both perception and reasoning based on actual model behaviors, and we found that most instances drawn from existing benchmarks did not meet this criterion. We then manually annotated all remaining artifacts to maintain tight control over correctness. While this level of manual curation is not scalable, we took care to maintain quality in this set.
>
> We fully agree that MathLens-General is too small to support standalone statistical conclusions. Its role in the paper is to serve as supplementary evidence, verifying that the trends observed in the geometry setting do not reverse when applied to non-geometry domains, rather than to draw strong general claims beyond geometry.
>
> ---
>
> ### **References**
>
> [1] Math-Verify (https://github.com/huggingface/Math-Verify)
>
> [2] Turpin, Miles, et al. "Language models don't always say what they think: Unfaithful explanations in chain-of-thought prompting." NeurIPS 2023.
>
> [3] MathVision, NeurIPS D&B 2024
>
> [4] EMMA, ICML 2025

---

### Official Review · Reviewer_q8XG · 2025-10-29

**Soundness:** 3
**Presentation:** 3
**Contribution:** 3
**Rating:** 6
**Confidence:** 2

**Summary:**

This paper introduces MATHLENS, a benchmark designed to disentangle the subskills of multimodal reasoning—specifically perception, reasoning, and their integration—in the context of geometry problems.
The authors argue that existing benchmarks often rely on aggregate accuracy, which obscures the specific capacities where models fail or improve.
MATHLENS is built from symbolic specifications to generate aligned annotations, including visual diagrams, textual descriptions, perception probes, and multimodal questions, ensuring consistency and robustness.
Through controlled experiments on open multimodal models, the paper reveals that different training approaches
have uneven effects.

**Strengths:**

1. MATHLENS provides a rigorous framework for decomposing multimodal reasoning into perception, reasoning, and integration, addressing a gap in existing evaluations. The use of symbolic semantics ensures controlled and reproducible annotations.
2. The paper thoroughly evaluates multiple model families across diverse settings, including robustness tests with semantic diagram modifications. This allows for nuanced insights into how training strategies affect specific capacities.
3. MATHLENS demonstrates strong alignment with datasets like MathVista and MathVerse, enhancing its credibility and utility for the community. The release of data and tools promotes reproducibility and further research.

**Weaknesses:**

See Questions.

**Questions:**

1. Have you explored preliminary strategies to improve integration? If so, what were the results? If not, what directions do you prioritize for future work?
2. MATHLENS relies on geometry problems, which have well-defined symbolic representations. For tasks with ambiguous symbolic mappings, how would you adapt MATHLENS’s decomposition framework to ensure consistent evaluation of perception, reasoning, and integration?
3. The diagram modifications test structural changes, but how would MATHLENS perform under real-world perturbations like occlusions or lighting variations?

---

> ### Author Response · Authors · 2025-11-17
>
> We thank the reviewer for recognizing the rigor of our decomposition framework, the benefits of symbolic semantics for controlled annotation, and the usefulness of our broad evaluations and released resources. We address the concerns below:
>
> ---
>
> ## **Question 1. Possible strategies for improving integration**
>
> We were unable to explore additional integration-targeted training strategies due to resource constraints. However, our findings suggest directions for improving integration.
>
> **(1) Integration is a distinct capacity.**
> Correctly combining visual facts with symbolic reasoning is qualitatively different from excelling at either component alone.
>
> **(2) Current architectures do not separate perception and reasoning during inference.**
> The model processes both in a single fused computation, which provides no internal structure for coordinating their interaction.
>
> **(3) This creates an inherent credit-assignment bottleneck.**
> When integration fails, the learning signal cannot tell whether the model misperceived, mis-reasoned, or failed at connecting the two.
>
> **(4) These observations highlight two promising directions for future work:**
>
> ### **(a) Grounded reasoning trajectories**
>
> Training data where the model must explicitly:
>
> - extract a visual fact,
> - ground it as an intermediate state, and
> - immediately use it in a reasoning step.
>
> Such grounded trajectories would give the model a clear structural signal for *when* and *how* perception should be invoked during reasoning.
>
> ### **(b) Process-level reward for credit assignment**
>
> Moving beyond answer-only supervision by rewarding:
>
> - correct extraction of intermediate visual facts,
> - correct incorporation of those facts into reasoning, and
> - correct propagation of grounded information.
>
> ---
>
> ## **Question 2. Annotation beyond symbolic domains**
>
> We collect and experiment with the **general-domain set (MathLens-General)** in Section 2.3 and Appendix C.2 and E.3. The set consists of reasoning on various domains as natural images or plots and charts.
>
> For domains lacking well-defined symbolic representations, we note that the only way to avoid data noise is **manual annotation with clear criteria**. We follow this strategy in building MathLens-General: manual but rule-based annotation to extract visual facts and convert them into perception probes. While this process does not scale as cleanly as geometry, the decomposition framework itself remains applicable as long as (i) visual facts can be enumerated, and (ii) they can be expressed as consistent probe questions. We will clarify this cross-domain adaptation in the revision.
>
> ---
>
> ## **Question 3. Photometric perturbations**
>
> Perturbations such as occlusion or lighting variation are meaningful primarily in natural-image settings. Our benchmark, however, focuses on **diagrammatic geometry**, where the task is fully determined by symbolic geometric relations. In this domain, **many appearance-level perturbations are not well defined** unless they preserve the underlying FormalGeo clause set; otherwise they alter the semantics of the problem rather than the model’s robustness.
>
> The MathLens decomposition framework can nevertheless accommodate additional perturbation types whenever the symbolic truth remains unchanged. In such cases, the pipeline would:
> (i) apply the perturbation to the diagram,
> (ii) keep the original clause set as the ground-truth semantics, and
> (iii) re-evaluate perception probes to check whether the model still recovers the required visual facts.
>
> While our study focuses on geometry-specific semantic modifications, the framework itself is compatible with broader perturbation families as long as they preserve the underlying structure. Expanding in that direction is therefore feasible but beyond the scope of the present work.

---

> > ### Comment · Reviewer_q8XG · 2025-11-26
> >
> > Thanks for the authors' response. I tend to maintain this positive score.

---

### Official Review · Reviewer_gFmh · 2025-10-31

**Soundness:** 3
**Presentation:** 3
**Contribution:** 3
**Rating:** 2
**Confidence:** 2

**Summary:**

The paper proposes a math benchmark to evaluate MLMs through different lenses rather than a single one. Specifically, they present and made publicly available a dataset called MathLens for geometry to perform evaluation based on three aspects: perception, reasoning and integration, enabling evaluation beyond a single score accuracy. Authors then did several fine-tuning analysis and made these main observations: (1) RL boosts perception, (2) textual SFT indirectly strengthens perception through reflective reasoning, (3) integration is the weakest skill among all, and finally (4) robustness varies (e.g., RL vs. SFT).

**Strengths:**

1. Paper is well-written and easy to follow. The core idea is interesting and novel.

2. A public release of the dataset for the community is nice and will help future research.

3. Extensive experiments and analyses were provided. For instance robustness analysis investigates changes to the diagram (e.g., rotation, etc.).

**Weaknesses:**

1. Single data source is just very limited -- only FormalGeo-7K was used as the basis of the data. This makes it hard to trust the findings IMO (e.g., relation between RL and SFT).

2. The core analysis were performed using open-weight models and it's unclear how the fine-tuning of closed-source models such as Gemini would make any difference.

3. Integration measurement is done indirectly -- this could be conflated with other latent failures, like context-length limitations.

**Questions:**

1. As mentioned in the paper, integration is the main bottleneck in structured geometry. Is this true for real-world less-structured data as well?

2. Since integration is the weakest skill, what training or architectural changes can be made to improve it?

---

> ### Author Response · Authors · 2025-11-17
>
> We appreciate the reviewer’s positive remarks on the clarity of the paper, the novelty of our core idea, the usefulness of releasing the dataset, and the breadth of our experimental and robustness analyses. We address the concerns below.
>
> ---
>
> ## **Weakness 1. Focus on geometry domain**
>
> First, **geometry is a canonical testbed for multimodal reasoning**. Leading benchmarks in this area [1-4] predominantly evaluate mathematical and geometric reasoning, with one dedicated entirely to geometry [2]. Mathematics is a foundational domain for textual reasoning [5], and geometric tasks provide a uniquely structured setting where perception, symbolic reasoning, and cross-modal integration can all be cleanly examined. Because the visual structure is precise and the symbolic semantics are well defined, geometry offers the tight experimental control required for decomposing multimodal reasoning. Focusing on geometry does not limit the value of our findings: rather, it aligns with how multimodal reasoning is typically evaluated.
>
> To complement this, **we also introduce MathLens-General**, detailed in Section 2.3 and Appendix C.2/E.3. This set covers **diverse natural-image and plot-based tasks**. The key findings, including the lack of integration improvement, persist in these general settings.
>
> ---
>
> ## **Weakness 2. Fine-tuning closed API models**
>
> Our goal is to understand how specific training regimes affect multimodal reasoning capacity. This requires knowing exactly what supervision a model has received. **Closed-source models provide no transparency about their fine-tuning objectives nor data mixture.** Fine-tuning them would therefore obscure (rather than illuminate) the causal relationship we aim to study.
>
> Moreover, **modern closed models are already heavily optimized for multimodal reasoning**. Additional fine-tuning with similar data typically produces little or no interpretable changes. As a result, fine-tuning closed models would not generate meaningful findings. We will clarify this rationale in the revised version.
>
> ---
>
> ## **Weakness 3. Controlling external variable**
>
> We controlled for variables that could confound integration measurement. All models use sufficiently large `max_token_length` to avoid truncation, and every “no-answer” output was re-run to eliminate transient issues such as network or API failures.
>
> Given these controls, the remaining source of error corresponds to the **compositional gap**: this conditional measurement of compositional failure is standard practice in the compositionality literature [6,7], and MathLens follows this established framing.
>
> ---
>
> ## **Question 1. Beyond geometry set**
>
> We perform the same automatic error decomposition on MathLens-General, our general-domain benchmark. Results in Figure 14 and Appendix E.3 show that integration does not improve consistently in natural-image and plot-based reasoning tasks either, mirroring the behavior observed on geometry.
>
> ---
>
> ## **Question 2. What current findings reveal about integration**
>
> Our analysis indicates why integration fails across training strategies and suggests where progress is needed.
>
> 1. **Integration is a distinct capacity.**
>    Combining visual facts with symbolic reasoning is fundamentally different from improving either component alone.
>
> 2. **Current architectures fuse perception and reasoning.**
>    They process both within a single computation, without any mechanism for coordinating their interaction.
>
> 3. **This creates a credit-assignment bottleneck.**
>    When an integration failure occurs, the model receives no signal identifying whether the issue lies in perception, reasoning, or the interface.
>
> These observations highlight two promising directions for future work:
>
> ### **(a) Grounded reasoning trajectories**
>
> Training data where the model must explicitly
>
> - extract a visual fact,
> - represent it as an intermediate grounded state, and
> - immediately use it within a reasoning step.
>
> This provides the structural signal that current models lack.
>
> ### **(b) Process-level reward for credit assignment**
>
> Instead of supervising only final answers, reward signals can target
>
> - correct extraction of visual facts,
> - correct incorporation into reasoning, and
> - consistent use of grounded information throughout the chain.
>
> ---
>
> ### **References**
>
> [1] OpenCompass Leaderboard (https://huggingface.co/spaces/opencompass/Open_LMM_Reasoning_Leaderboard)
>
> [2] MathVerse, ECCV 2024 (https://arxiv.org/pdf/2403.14624)
>
> [3] MathVision, NeurIPS D&B 2024 (https://arxiv.org/pdf/2402.14804)
>
> [4] MathVista, ICLR 2024 (https://arxiv.org/pdf/2310.02255)
>
> [5] AIME 2025 (https://huggingface.co/datasets/MathArena/aime_2025)
>
> [6] “Measuring and Narrowing the Compositionality Gap in Language Models”, EMNLP Findings 2023 (https://arxiv.org/pdf/2210.03350)
>
> [7] “Does my multimodal model learn cross-modal interactions? It’s harder to tell than you might think!”, EMNLP 2020 (https://arxiv.org/pdf/2010.06572)

---

> > ### Comment · Reviewer_gFmh · 2025-11-17
> >
> > I'd like to thank authors for addressing all my concerns and am happy to raise my score.

---

### Author Response · Authors · 2025-12-01
**Summary of Review Process and Outstanding Issues**

To facilitate assessment of the review process, we summarize the overall reviewer landscape and our responses to the remaining concerns.

---

## **Main contribution of the paper**

- Introduces **MathLens**, a benchmark that disentangles multimodal reasoning into perception, reasoning, and integration, addressing limitations of aggregate-accuracy evaluations.
- Constructs diagrams, textual descriptions, perception probes, and multimodal questions from **symbolic geometric specifications**, ensuring aligned multimodal annotations.
- Provides **controlled analyses** across multiple multimodal training regimes, showing uneven improvements across the three capacities and identifying integration as the central bottleneck.
- Describes the benchmark, evaluation code, data-generation pipeline, and experimental artifacts, which we are preparing for public release.

---

## **Overall reviewer stance.**

**Three of four reviewers (gFmh, q8XG, sN3K) expressed positive evaluations and explicitly supported acceptance after the rebuttal. Reviewer gFmh raised the score from 2 to 6, noting that all concerns were addressed.** These reviewers highlighted the clarity of the framework, the novelty of the decomposition, and the diagnostic value of the analyses.

---

## **Primary remaining concern.**

Reviewer atqw maintained a borderline-negative score, focusing on statistical methodology, evaluation transparency, and generalizability.
- **Statistical significance.** We clarified that evaluation is deterministic (exact-match, no stochastic decoding), eliminating run-to-run noise. Furthermore, with a test set of **N=926 unique problems** and **N=6,510 visual probes**, the sample sizes are sufficiently large to ensure that the reported performance gaps are statistically robust and distinct from sampling variance. The variance in Figure 8 reflects systematic robustness conditions, not evaluation noise. We will report any remaining variance and apply tests where appropriate.


- **Evaluation procedure transparency.** We clarified exact-match scoring for perception and reasoning and will provide evaluation scripts.


- **Human validation.** We clarified that human annotation is **not a gold label** for categorizing failure types, as Chain-of-Thought traces do not guarantee faithful reflection of the model's causal computation. Consequently, humans cannot reliably infer the true source of an error from the output alone. Nonetheless, we conducted a human review following established protocols, finding strong agreement with our automated scoring for perception and reasoning.


- **Domain limitation.** We clarified why geometry remains a standard and controlled setting for multimodal reasoning, and we supplemented this with **MathLens-General** to show that the observed trends do not reverse in non-geometry domains.


---

## **Context for remaining disagreement.**

Reviewer atqw highlighted a preference for statistical testing even under deterministic evaluation, as well as interest in human validation for categories that are not straightforward for human annotators to assess reliably. These concerns relate to broader methodological expectations rather than unresolved issues with the paper’s technical content. In contrast, the other three reviewers indicated that the clarifications fully addressed their earlier questions and maintained positive evaluations.

---

We are happy to provide additional details if they would assist the AC’s assessment.

---

### Meta-Review · Area_Chair_MvsA · 2026-01-12

**Summary:**

The paper proposes MathLens, a benchmark designed to disentangle multimodal reasoning skills in LLMs into perception, reasoning, and integration, primarily using geometry problems. The authors utilize symbolic specifications from the FormalGeo dataset to generate aligned annotations and probes. The submission reports that different training strategies (RL vs. SFT) impact these sub-skills differently, identifying "integration" as a primary bottleneck.

While reviewers gFmh, q8XG, and sN3K were generally positive , reviewer atqw raised significant concerns regarding methodological rigor. Specifically, atqw questioned 1) the statistical validity of the comparative claims (e.g., "RL improves perception") in the absence of significance testing or confidence intervals, 2) the assumption that the perception probes are truly exhaustive enough to isolate "integration" as a distinct residual failure mode rather than a perceptual one and 3) the domain limitation (MathLens-General is indeed too small as a benchmark). Moreover, the promised change should be reflected in the manuscript rather than remained as a promise.

**Reviewer Concerns:**

Addresses Concerns:
- Closed-Source Models (gFmh): The authors clarified that fine-tuning closed-source models is infeasible for causal analysis due to opacity regarding their training data. gFmh found this response satisfactory.

- Probe Integrity (sN3K): The authors explained that perception probes are derived from formal clause sets in FormalGeo, ensuring theoretical completeness. sN3K was satisfied with this explanation.

Outstanding Concerns:
- Statistical Significance (atqw): This remains the critical unresolved issue. Reviewer atqw argued that the authors' comparative claims (e.g., distinguishing the effects of RL vs. SFT) require statistical validation. The authors argued that because their decoding is deterministic (greedy), statistical variance is non-existent. atqw strongly rebutted this, noting that deterministic decoding does not eliminate the need to validate that performance differences across a sample set are statistically meaningful rather than artifacts of the specific problem selection.

- Definition of Integration (atqw): atqw remained unsatisfied about the fact that "integration" is cleanly disentangled. If a perception probe is not perfectly exhaustive, errors classified as "integration" may simply be subtle perceptual failures. The reviewer noted that the "integration" metric might be confounding different types of failures, reducing the reliability of the paper's central claims.

- Generalizability and MathLens-General Size (atqw): Reviewer atqw noted that the primary analysis is "derived exclusively from the geometry domain," limiting generalizability. While the authors pointed to a supplementary "MathLens-General" set, the limited size and scope of this secondary benchmark are insufficient to dismiss the concern.

**Reviewer Scores:**

- Reviewer gFmh: 2 to 6 (The reviewer explicitly raised their score from 2 to 6 after the rebuttal addressed their concerns).
- Reviewer q8XG: 6 (The reviewer maintained their positive assessment, valuing the rigorous framework and semantic diagram modifications ).
- Reviewer sN3K: 6 (The reviewer voted in favor of acceptance).
- Reviewer atqw: 4 (The reviewer maintained a negative stance, stating that the paper "makes strong empirical claims but lacks necessary methodological rigor" and that their concerns remained unaddressed despite the author response )


While the concept of disentangling perception from reasoning in MLLMs is a timely pursuit, the current submission falls short of the methodological standards required for ICLR. The primary reason for rejection is the lack of statistical and methodological rigor and the study is restricted to the geometry domain, and the "MathLens-General" benchmark is too small to prove that the identified "integration bottleneck" exists across broader multimodal tasks or even only mathematical tasks.It is important to note that while two reviewers provided positive scores (6) and one raised his/her score to 6, these scores were not accompanied by high confidence or strong, proactive support for the paper's acceptance during the discussion phase. Given these cumulative concerns, the paper is not recommended for acceptance.

---

### Decision · Program_Chairs · 2026-01-26

Reject